# Good SFT Optimizes for SFT, Better SFT Prepares for Reinforcement Learning

**Dylan Zhang** [1]  **Yufeng (Felix) Xu** [1 2]  **Haojin Wang** [1]  **Qingzhi Chen** [1]  **Hao Peng** [1]

## Abstract

Post-training of reasoning LLMs is a holistic process that typically consists of an offline SFT stage followed by an online reinforcement learning (RL) stage. However, SFT is often optimized in isolation to maximize SFT performance alone. We show that, after identical RL training, models initialized from stronger SFT checkpoints can significantly underperform those initialized from weaker ones. We attribute this to a mismatch typical in current SFT–RL pipelines: the distribution that generates the offline SFT data can differ substantially from the policy optimized during online RL, which learns from its own rollouts. We propose PEAR (**P**olicy **E**valuation–inspired **A**lgorithm for Offline Learning Loss **R**eweighting), an SFT-stage method that corrects this mismatch and better prepares the model for RL. PEAR uses importance sampling to reweight the SFT loss, with three variants operating at the token, block, and sequence levels. It can be used to augment standard SFT objectives and incurs little additional training overhead once probabilities for the offline data are collected. We conduct controlled experiments on verifiable reasoning games and mathematical reasoning tasks on Qwen2.5/3 and DeepSeek-distilled models. PEAR consistently improves post-RL performance over canonical SFT, with Pass@8 gains up to 30 percentage points on AIME-2025. Our results suggest that PEAR is an effective step toward more holistic LLM post-training by designing and evaluating SFT with downstream RL in mind rather than in isolation.

## 1. Introduction

Post-training of reasoning language models typically follows a two-stage paradigm: an *offline* supervised fine-tuning (SFT) phase produces an initial checkpoint, which is then used to initialize an *online* reinforcement learning (RL) phase that further enhances the model (Shao et al., 2024; Guo et al., 2025; Yang et al., 2024). Both areas have become active research fronts. In particular, a growing body of work has proposed offline learning objectives to improve SFT, often by reweighting or regularizing next-token likelihood (Qin & Springenberg, 2025; Zhu et al., 2025c; Wu et al., 2025; Lin et al., 2025; Li et al., 2025a).

From a practical perspective, the performance of interest is usually the model's final accuracy after completing both SFT and downstream RL. However, it is common that these existing techniques optimize for SFT-stage performance in isolation, often with the implicit assumption that gains in offline performance will translate to improved performance after RL. Kang et al. (2025) show that repetition and data homogeneity boost SFT but may reduce RL headroom. This motivates us to investigate if offline gains of an objective could also be a misleading proxy for its effectiveness as an RL initialization. We empirically show the gains of a stronger offline checkpoint over a weaker one can shrink, disappear, or even reverse after both undergo identical RL training. Therefore, optimizing for offline performance alone may be counterproductive when the goal is strong final performance after RL (Fig. 1 in §2).

We contend that the goal of an offline stage is not merely strong offline accuracy, but an initialization that facilitates improvement under the online RL. This requires addressing a distribution mismatch between offline and online stages: Typically, during SFT, the model learns from data sampled from a different distribution, often dubbed the **behavior policy** (Sutton & Barto, 2018; Precup et al., 2000; Uehara et al., 2022). In contrast, during online RL, the target of learning (thereby the **target policy**), learns from roll-outs generated by itself. There is a clear distribution mismatch that needs to be corrected between them (Zhao et al., 2022; Lee et al., 2021; Zu et al., 2025) in order for an effective offline-to-online transition.

It is therefore crucial to quantify and correct this distribution mismatch. Inspired by off-policy evaluation (OPE) (Precup

[1]University of Illinois Urbana-Champaign [2]New York University (Shanghai). Work done during internship at UIUC. Correspondence to: Dylan Zhang <shizhuo2@illinois.edu>, Hao Peng <haopeng@illinois.edu>.

*Proceedings of the 43rd International Conference on Machine Learning*, Seoul, South Korea. PMLR 306, 2026. Copyright 2026 by the author(s).

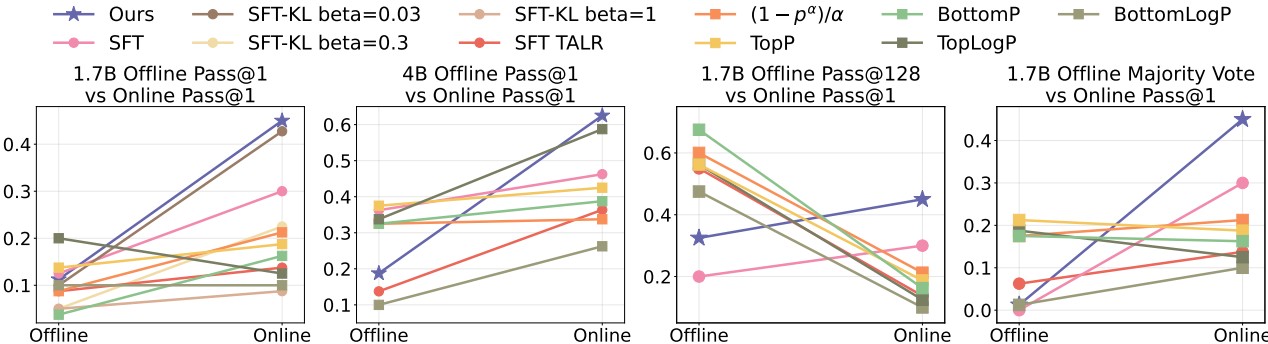

*Figure 1.* Offline v.s. Online pass@1 on SynLogic Games on a total of 19 Models. It exhibits significant ranking changes indicating offline performance will not entail online performance. In addition, our proposed approach remains the most effective initialization for online RL. Panels (b)–(d) show that this rank instability is not specific to offline Pass@1: stronger offline Pass@8, Pass@128 (large-$K$ coverage), and majority-vote accuracy also do not reliably predict online Pass@1.

et al., 2000; Thomas & Brunskill, 2016; Jiang & Li, 2016; Levine et al., 2020), we address this by reweighting offline data using importance weights, i.e., the likelihood ratio between the target policy and the behavior policy (§3). Intuitively, this reweighting scales each token's loss according to how likely the target policy would generate the same continuation relative to the behavior policy, so that offline training better reflects the trajectories that online RL will actually revisit. We present a sequence-level (Rowland et al., 2020) and a token-wise reweighting based on suffix-ratios (Precup et al., 2000). We also present variants that improve stability by block-wise weighing and leveraging negative data.

We evaluate PEAR and its variants on reasoning games and math benchmarks across 6 different models of various sizes: Qwen3-Base-0.6B; 1.7B; 4B and 8B (Yang et al., 2025), Qwen2.5-1.5B-Math (Yang et al., 2024) and DeepSeek-Distill-Qwen-1.5B (Guo et al., 2025). Using an SFT–RL pipeline that varies only the SFT-stage objectives, PEAR and its variants consistently improve *post-RL* performance over strong baselines. Comparing checkpoints finetuned using PEAR versus canonical SFT on the same data, the former outperforms the latter by up to +25 pp absolute Pass@1 on SynLogic logic games (Qwen3-1.7B-Base, $13.1\% \to 38.3\%$; Table 4) and up to +30 pp Pass@8 on AIME-25 (Balunović et al., 2026) (DS-Qwen-1.5B, $5\% \to 35\%$; Table 2). Moreover, our analysis shows that PEAR-initialized models undergo less parameter drift during RL (§4.7). The takeaway is simple: *a good offline stage should not optimize for offline accuracy, but prepare the policy for the RL that follows*. PEAR is one concrete instantiation of this principle.

## 2. Offline Performance May Not Entail Online

There are various techniques to improve supervised finetuning (SFT) for reasoning, typically targeting stronger *offline* performance or reduced forgetting.

Recent reasoning LM post-training pipeline typically applies an online RL stage to further improve performance after SFT (Shao et al., 2024; Guo et al., 2025; Yang et al., 2024). In this setting, the offline stage provides the initialization for RL, and prior study (Kang et al., 2025) has identified that dataset construction and hyper-parameter affects SFT and RL performances differently in this pipeline. This naturally leads us to a question:

> Will the advantage of an offline learning objective carry over to post-RL performance?

We experiment with a wide spectrum of objectives covering the span the standard SFT "loss-strength" spectrum: drift control (KL), smooth probability-shaped reweighting and hard masking toward high-/low-confidence tokens.

Li et al. (2025a) presents a generalized view of SFT loss by studying a series of probability-based objectives (Table 1), where one could control how strongly training emphasizes low- vs. high-probability tokens by altering the transformation of probability, and learning selectively from easy / difficult tokens. TALR similarly modifies SFT via adaptive token-wise reweighting (Table 1). We follow their recommended hyperparameters; see Appendix K.2 and K.1.

In addition, we consider standard negative log-likelihood (NLL) loss; and KL-regularized NLL $\mathcal{L}(\theta) = \mathbb{E}_{(x,y)\sim\mathcal{D}}\big[ -\log \pi_\theta(y \mid x)\big] + \beta\,\mathbb{E}_{x\sim\mathcal{D}}[\mathrm{KL}(\pi_\theta(\cdot \mid x) \,\|\, \pi_{\mathrm{ref}}(\cdot \mid x))]$. We alter $\beta \in \{0.03, 0.1, 0.3, 1\}$ for KL-regularized variant.

We perform a controlled, contamination-free experiment by applying each of these offline objectives followed by online RL on synthetic logic puzzles from (Liu et al., 2025).

### 2.1. Offline $\neq$ Online

Figure 1 visualizes offline versus online performance. While several objectives indeed outperform SFT offline on Pass@1,

| Name | Per-token objective / weight |
|------|------------------------------|
| SFT (NLL) | $\ell(p) = -\log p$ |
| SFT+KL | $\ell(p) = -\log p + \beta\,\mathrm{KL}$ |
| GeneralFamily-$\alpha$ | $\ell_\alpha(p) = \dfrac{1-p^\alpha}{\alpha}$ $(\alpha \to 0 \Rightarrow -\log p)$ |
| TopP-$q$ | $(1-p)\,\mathbf{1}[p \geq q]$ |
| BottomP-$q$ | $(1-p)\,\mathbf{1}[p \leq q]$ |
| TopLogP-$q$ | $-\log(p)\,\mathbf{1}[p \geq q]$ |
| BottomLogP-$q$ | $-\log(p)\,\mathbf{1}[p \leq q]$ |
| TALR | $w_t \propto \exp(-\ell_t/\tau) = p_t^{1/\tau}$ |
| SFT+KL | $-\log(p) + \beta \mathrm{D_{KL}}$ |

*Table 1.* Compared objectives at the per-token level. Here $p_t := p_\theta(y_t \mid y_{<t}, x)$.

these gains do not reliably translate to stronger post-RL models: some checkpoints are simply harder for subsequent RL to improve and ultimately lose their offline advantage. One may be tempted to pick **TopLogP** for Qwen3-1.7B-Base because of the best offline scores, yet this choice leads to worst-among-all post-RL performance, even under-performing SFT initialized model.

We inspect other descriptors for sampling that potentially relates to RL: offline pass@K with large K (Yue et al., 2025) and majority voting accuracy (Kang et al., 2025) may correlate with the RL performance, we show (in Figure 1) that the ranking is not always well-preserved when comparing the effectiveness of different techniques either.

### 2.2. Why Uniform Loss Is Misaligned

Standard SFT and KL-distillation apply uniform token-level supervision under prefixes induced by the behavior (data-generating) policy $\pi_\beta$, while online RL samples and optimizes rollouts from the evolving target policy $\pi_\theta$. This behavior–target occupancy mismatch—well known in offline-to-online RL—can hurt the subsequent online phase (Huang et al., 2025; Zu et al., 2025; Lee et al., 2021; Zhao et al., 2022).

In auto-regressive generation, small early mismatches shift the prefix distribution and propagate forward, compounding over long horizons (Ross et al., 2011; Mehta et al., 2024; Liu et al., 2019; Ross & Bagnell, 2014; Sun et al., 2017). This is especially acute for long-form reasoning (Guo et al., 2025), where traces often involve implicit search (trial, backtracking, self-correction): $\pi_\beta$ may over-represent continuations that are effectively dead-ends under $\pi_\theta$, so uniformly training on all logged tokens can reinforce transitions that RL will rarely revisit (Fig. 2).

### 2.3. Off-Policy Evaluation

To reason about the offline-to-online mismatch, we adopt an off-policy evaluation (OPE) lens: we have logged trajectories from a *behavior* policy $\pi_\beta$, while the subsequent

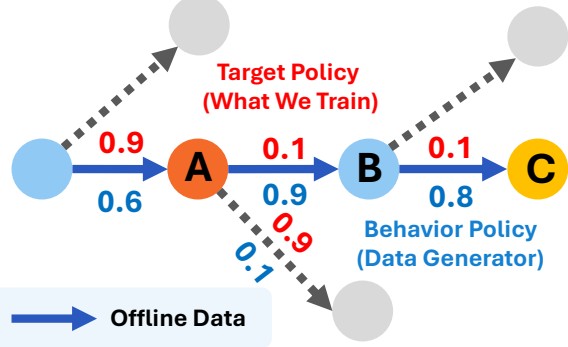

*Figure 2.* A sketch of our weighing intuition. **Red** numbers are probabilities under target policy, **Blue** numbers are probabilities under behavior policy. After token $A$, the behavior (data-generating) policy often continues with $A \to B \to C$ (e.g., $0.9 \times 0.8$), but this continuation is highly unlikely for the policy we ultimately want to optimize. As a result, the offline data over-represents $A \to B \to C$, which can push the model to associate $A$ with an implausible continuation. During online RL, once the model generates $A$, it will rarely follow with $B$ and $C$, so learning from these offline continuations provides little useful signal. We therefore down-weight token $A$ to avoid visiting it.

online RL stage generates rollouts from a (changing) *target* policy $\pi_\theta$. Classical OPE corrects this behavior–target shift via a change of measure with likelihood ratios (Precup et al., 2000; Jiang & Li, 2016; Uehara et al., 2022; Levine et al., 2020): $\mathbb{E}_{\tau \sim \pi_\theta}[f(\tau)] = \mathbb{E}_{\tau \sim \pi_\beta}\left[\frac{\pi_\theta(\tau)}{\pi_\beta(\tau)} f(\tau)\right]$.

OPE comprises a family of estimators that correct for behavior–target mismatch using likelihood ratio, including variants that compute likelihood ratio across the entire trajectory $w = \prod_{t=1}^{T} \frac{\pi_\theta(a_t|s_t)}{\pi_\beta(a_t|s_t)}$ (Thomas & Brunskill, 2016) and uniformly apply to all actions for simplicity (Rowland et al., 2020) and those that compute suffix ratios from a certain time step $w_n = \prod_{t=n+1}^{T} \frac{\pi_\theta(a_t|s_t)}{\pi_\beta(a_t|s_t)}$ for each decision (Precup et al., 2000). This naturally suggests using likelihood-ratio-based sequence or continuation weights as compatibility signals between the logged data and the current policy to correct for the mismatch mentioned in §2.2 with different granularity.

> **TAKEAWAYS**
>
> **Offline $\neq$ online:** better algorithm in offline scores need not yield better post-RL performance.

### 3. Method

To address the offline-to-online mismatch identified in § 2, we introduce PEAR (**P**olicy **E**valuation–inspired **A**lgorithm for Offline Learning Loss **R**eweighting), a reweighting scheme for offline fine-tuning that produces a stronger initialization for subsequent online RL on verifiable reasoning. PEAR keeps the underlying objective unchanged (SFT or

KL-based distillation) and modifies only how each token's loss is weighted.

---

**APPROACH SUMMARY**

**Step 1.** Compute token log-likelihood ratios on tokens from offline dataset.
**Step 2.** Aggregate into weights using either one of 3 variants and stabilize it.
**Step 3.** Weigh the loss for each token.

---

### 3.1. Problem setup and notation

We consider standard offline fine-tuning from a dataset of prompt–response pairs. $\mathcal{D} = \{(x, \mathbf{y})\}$, where $x$ is a prompt and $\mathbf{y}$ is a token sequence produced by a known data-generating policy $\pi_\beta$. We train a target model $\pi_\theta$ on $\mathcal{D}$. Let $\pi_\theta(y_t \mid x, \mathbf{y}_{<t})$ denote the model probability at position $t$, and let $\ell_\theta(x, \mathbf{y}_{<t}, y_t)$ be a per-token training loss.

### 3.2. PEAR-weighted Offline Training Objective

PEAR comes as a simple weighting approach for standard offline loss on each token. Given an example $(x, \mathbf{y}) \sim \mathcal{D}$ from the dataset, we first compute the weight $G_t$ following either one of the 3 weighting strategies we will present below. We will apply numerical stabilization techniques discussed in a subsequent subsection §3.7 and denote the numerically-stabilized version as $\hat{G}_t$. PEAR keeps the underlying per-token loss $\ell_\theta(x, \mathbf{y}_{<t}, y_t)$ unchanged (SFT/NLL or KD/forward-KL), and only reweighs it:

$$\mathcal{L}_{\text{PEAR}}(\theta) \triangleq \mathbb{E}_{(x,\mathbf{y})\sim\mathcal{D}} \left[ \sum_{t=1}^{T} \text{sg}\left[ \widehat{G}_t \right] \ell_\theta(x, \mathbf{y}_{<t}, y_t) \right],$$

where $\text{sg}[\cdot]$ stops gradients through the weights (we treat $\widehat{G}_k$ as a fixed coefficient, not an additional differentiable path).

### 3.3. Sequence-Level Weighting

We start by presenting the simplest form of PEAR weighting: sequence-level importance weighting. For each token, let us define $\Delta_t \triangleq \frac{\pi_\theta(y_t|x, y_{<t})}{\pi_\beta(y_t|x, y_{<t})}$, to denote the probability ratio between policy we want to train $\pi_\theta$ and the behavior (data-generating) policy $\pi_\beta$. Then, the resulting sequence-level importance ratio is $w_{1:T} \triangleq \frac{\pi_\theta(\mathbf{y}|x)}{\pi_\beta(\mathbf{y}|x)} = \prod_{t=1}^{T} \Delta_t$ can be used to represent sequence's relative likelihood under $\pi_\theta$ to $\pi_\beta$. This allows us to estimate the loss under the target policy's distribution: $\mathbb{E}_{y\sim\pi_\theta(\cdot|x)}[\ell_\theta(x, y)] = \mathbb{E}_{y\sim\pi_\beta(\cdot|x)}[w_{1:T} \ell_\theta(x, y)]$. We therefore use this weight $G_i \triangleq w_{1:T} \quad \forall i = 1, 2, ..T$ to equally weigh each token in the trajectory. In our experiments, we show that this simple weighting mechanism can yield strong performance.

---

**Algorithm 1** PEAR

**Require:** One example $(x, y)$ with $y = (y_1, \ldots, y_T)$; model $\pi_\theta$; behavior policy $\pi_\beta$.
**Require:** Block size $B$; mode `uniform` or `suffix`; discount $\gamma \in (0, 1]$.
**Require:** Token loss $\ell_\theta(x, y_{<t}, y_t)$; clip bounds $[\ell_\Delta, u_\Delta]$, $[G_{\min}, G_{\max}]$.
**Ensure:** Weighted loss $L(\theta)$.
1 Partition $\{1, \ldots, T\}$ into $K = \lceil T/B \rceil$ contiguous blocks $\{\mathcal{I}_k\}_{k=1}^{K}$;     ▷ token-level weighting: $B = 1$
    let $e_k = \max \mathcal{I}_k$.
2 **Per-token quantities.**
3 $\delta_t \leftarrow \text{clip}(\log \pi_\theta(y_t \mid x, y_{<t}) - \log \pi_\beta(y_t \mid x, y_{<t}), \ell_\Delta, u_\Delta)$, $\forall t \in [T]$
         ▷ clipped log-ratios for numerical stability
4 $\ell_t \leftarrow \ell_\theta(x, y_{<t}, y_t), \quad \forall t \in [T]$
         ▷ token loss under the base objective (SFT/KD)
5 **Blockwise reductions.**
6 $\rho_k \leftarrow \sum_{t\in\mathcal{I}_k} \delta_t, \quad \forall k \in [K]$
    ▷ block log-ratio (log of within-block product of $\pi_\theta/\pi_\beta$)
7 $b_k \leftarrow \sum_{t\in\mathcal{I}_k} \ell_t, \quad \forall k \in [K]$
         ▷ aggregate loss for this block
8 **if** `uniform` **then**
9    $\widehat{G}_T \leftarrow \text{clip}\left(\exp\left(\sum_{k=1}^{K} \rho_k\right), G_{\min}, G_{\max}\right)$
10    $\widehat{G}_t \leftarrow \widehat{G}_T, \quad \forall t \in [T]$
11    **return** $\sum_{k=1}^{K} \text{sg}\left[\widehat{G}_T\right] b_k$   ▷ equivalently $\sum_{t=1}^{T} \text{sg}\left[\widehat{G}_t\right] \ell_t$
12 **end if**
13 **Suffix mode: continuation weights (single backward scan).**
14 $L \leftarrow 0; u \leftarrow 0$     ▷ $u$ tracks future log-ratio: $\sum_{m=k+1}^{K} \rho_m$
15 **for** $k = K$ downto 1 **do**
16    $\widehat{G}_{e_k} \leftarrow \text{clip}\left(\exp\left((T - e_k)\log\gamma + u\right), G_{\min}, G_{\max}\right)$
     ▷ weight by how plausible the *remaining continuation* is under $\pi_\theta$
17    $\widehat{G}_t \leftarrow \widehat{G}_{e_k}, \quad \forall t \in \mathcal{I}_k$     ▷ same $G_t$ within a block
18    $L \leftarrow L + \text{sg}\left[\widehat{G}_{e_k}\right] b_k$
19    $u \leftarrow u + \rho_k$
20 **end for**
21 **return** $L$

---

### 3.4. Token-level Weighting Based on Continuation

We then take a more granular view on the sequence. Sequence-level PEAR uniformly applies the global importance score to each token, yet the weights may not be uniform across positions, a sequence may become 'unlikely" because of implausible regions. For a token $y_t$, we evaluate whether the *continuation from dataset* $\mathbf{y}_{>t}$ remains plausible under the model *conditioned on* taking $y_t$. If the relative plausibility is small, it means gradients at time $t$ primarily encourage tokens that lead into regions that $\pi_\theta$ is unlikely to revisit when sampling from itself. We therefore downweight the loss on such tokens, focusing offline updates on prefixes whose continuations from the dataset are compatible with the current policy. To this effect, we introduce a token-level importance weighting based on the suffix importance ratio, where $G_t = \gamma^{T-t} \prod_{j=t+1}^{T} \Delta_j$ where $\gamma \in (0, 1]$

is a discount factor to control variance in long horizon (Sutton & Barto, 2018; Jiang & Li, 2016).

### 3.5. Block-Level Weighting to Improve Stability

A product over long horizons inevitably introduce large variance (Bossens & Thomas, 2024; Liu et al., 2018; 2020). To reduce the effective length of the multiplicative importance-weight for better stability, we present a block-level variant that trades granularity for stability.

We partition positions $\{1, \ldots, T\}$ into $K = \lceil T/B \rceil$ contiguous blocks $\{\mathcal{I}_k\}_{k=1}^K$, each of length at most $B$. Let $e_k \triangleq \max \mathcal{I}_k$ denote the last index of block $k$.

For each block $k$, define the product of token-level ratios within the block as $\Delta_k^{\text{blk}} \triangleq \prod_{t \in \mathcal{I}_k} \Delta_t$.

Let $S_k \triangleq \prod_{j=e_k+1}^T \Delta_j$ denote the importance ratio of the suffix after block $k$ (i.e., how likely the remaining continuation is under $\pi_\theta$ relative to $\pi_\beta$). Equivalently, $S_k$ can be computed block-wise as $S_k = \prod_{m=k+1}^K \Delta_m^{\text{blk}}$.

We assign every token in block $k$ the same discounted continuation weight $G_t = G_k^{\text{blk}} \triangleq \gamma^{T-e_k} S_k, \forall t \in \mathcal{I}_k$.

Note that when $B = 1$, we recover the token-level PEAR introduced in §3.4.

### 3.6. Optionally Incorporating Negative Examples

When $\mathcal{D}$ contains verified failures, we optionally add a repulsive term that discourages imitating negative trajectories in a policy-consistent way. Let $\mathcal{D}^- = \{x, \mathbf{y}^-\}$ denote failures. We can still compute sequence-level weights and apply a repulsive term on those data points:

$$\mathcal{L}_{\text{neg}}(\theta) \triangleq \mathbb{E}_{(x,\mathbf{y}^-)\sim\mathcal{D}^-}\left[-\lambda\,\text{sg}\left[\widehat{G_t^-}\right]\sum_{t=1}^T \ell_\theta(x, \mathbf{y}_{<t}^-, y_t^-)\right],$$

where $\widehat{G_t^-}$ is a sequence level weight on negative trajectories. Here, we perform gradient ascent to push the model away from the negative response with a trajectory-level weight.

### 3.7. Numerical Stabilization

For numerical stability, we compute importance weights in log-space to avoid products of ratios over long sequences. We apply clip on both per-decision ratios $\Delta_t$ and final weights $\hat{G}_t$ as described in Algorithm 1.

## 4. Experiments

We present a careful controlled study under clean set-ups to study the effectiveness of various PEAR-based weighting, that in turn proves the insights in § 2. Our main experiments target *verifiable reasoning* (math and logic games),

where rule-based verifiers eliminate reward confounds; we additionally verify that PEAR remains effective beyond this regime — on instruction-following tasks (App. D), with proxy/ensemble behavior policies when $\pi_\beta$ is not directly accessible (App. A), and paired with DAPO instead of GRPO (App. G).

### 4.1. Tasks and data

#### 4.1.1. LOGIC GAMES

**Task Sources.** We use synthetic, verifiable puzzles from SynLogic (Liu et al., 2025) and Enigmata (Chen et al., 2025b) as a primary testbed. Both of them are synthetic reasoning environments that procedurally generate verifiable puzzle instances from diverse environments to allow noise-free data collation, training and evaluation. This allows for a minimally confounded evaluation setting, with reduced exposure to knowledge dependence and contamination.

**Offline buffer construction.** We generate synthetic games using the rule-based generator and de-duplicate prompts across train/test and remove any train instances that overlap with evaluation prompts. We sample responses with Qwen3-8B (Yang et al., 2025) and verify final answers. The resulting offline buffer contains roughly 100,000 correct trajectories.

**Evaluation.** We measure Pass@$\{1, 8\}$ on a held-out set of puzzles with the original verifiers. We evaluate using samples from SynLogic's evaluation set.

#### 4.1.2. MATH REASONING

**Data.** For offline training, we use the subset of all math problems in SYNTHETIC-2 dataset (Prime Intellect, 2025) – a total of 33,400 unique instructions. We sample responses from Qwen3-8B and verify with final answer, forming a dataset of 100,000 question-response pairs. For online RL, we use DAPO-17k dataset (Yu et al., 2025).

**Evaluation.** Following common practice, we evaluate on MATH-500 (Hendrycks et al., 2021), MINERVA (Lewkowycz et al., 2022), AIME-2024, AIME-2025 and AMC-2023 (Balunović et al., 2026). We report average accuracy across 64 samples to reduce variance and pass@K.

### 4.2. Training details

**Offline training** Unless stated otherwise, we train for 1 epoch with learning rate $3 \times 10^{-5}$ on games and $1 \times 10^{-5}$ on math. For PEAR, we use $\gamma = 0.999$, clip $\log \hat{G}_t$ to $[-10, 5]$, and clip per-decision $\log \Delta_t$ to $[-0.08, 0.3]$.

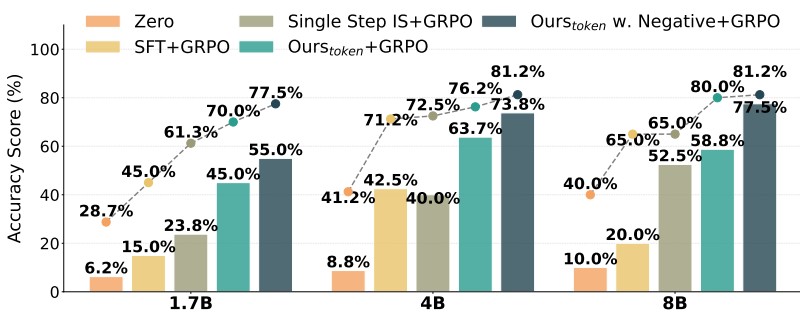

*(a)* Pass@1 comparison across initializations for SynLogic Games. PEAR$_{B=1}$ significantly improve upon SFT initialization, and incorporating negative gradients can further improve Pass@1.

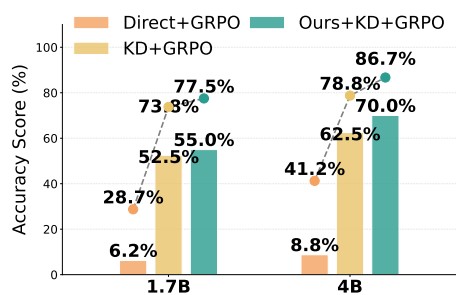

*(b)* PEAR$_{B=1}$ applied to KL based knowledge distillation.

*Figure 3.* Results on SynLogic dataset. We demonstrate that PEAR consistently improves post-RL performance. The bars reflect Pass@1 and dots mark Pass@8. In Figure 3a, Single Step IS is a baseline that corrects each token only based on the probability ratio of the token itself. See §4.3.

| Base Model | | QWEN2.5-1.5B-MATH | | DS-QWEN-1.5B | | QWEN3-4B-BASE | | QWEN3-8B-BASE | |
|---|---|---|---|---|---|---|---|---|---|
| | | SFT +GRPO | PEAR$_{B=1}$ +GRPO | SFT +GRPO | PEAR$_{B=1}$ +GRPO | SFT +GRPO | PEAR$_{B=1}$ +GRPO | SFT +GRPO | PEAR$_{B=1}$ +GRPO |
| **BENCHMARK** | **PASS@** | | | | | | | | |
| **AIME24** | AVG. 64 | 5% | 8% (+3) | 1% | 14% (+13) | 13% | 15% (+2) | 15% | 19% (+4) |
| | **8** | 19% | 23% (+4) | 2% | 38% (+36) | 25% | 40% (+15) | 35% | 41% (+6) |
| | **64** | 37% | 47% (+10) | 7% | 57% (+50) | 43% | 60% (+17) | 60% | 67% (+7) |
| **AIME25** | AVG. 64 | 3% | 8% (+5) | 1% | 14% (+13) | 7% | 15% (+8) | 17% | 18% (+1) |
| | **8** | 14% | 24% (+10) | 5% | 35% (+30) | 21% | 35% (+14) | 35% | 35% (0) |
| | **64** | 30% | 40% (+10) | 17% | 53% (+36) | 40% | 57% (+17) | 53% | 53% (0) |
| **AMC23** | AVG. 64 | 42% | 47% (+5) | 20% | 56% (+36) | 54% | 59% (+5) | 45% | 55% (+10) |
| | **8** | 80% | 78% (-2) | 50% | 91% (+41) | 83% | 88% (+5) | 75% | 85% (+10) |
| | **64** | 98% | 98% (0) | 63% | 98% (+35) | 93% | 95% (+2) | 95% | 98% (+3) |
| **Olympiad** | AVG. 64 | 29% | 34% (+5) | 13% | 41% (+28) | 41% | 46% (+5) | 32% | 40% (+8) |
| | **8** | 55% | 57% (+2) | 34% | 66% (+32) | 60% | 66% (+6) | 60% | 65% (+5) |
| **MATH500** | AVG. 64 | 63% | 70% (+7) | 36% | 72% (+36) | 78% | 80% (+2) | 66% | 74% (+8) |
| | **8** | 89% | 91% (+2) | 68% | 94% (+26) | 93% | 93% (0) | 90% | 93% (+3) |
| **Pass@1 Avg.** | | 28% | 33% (+5) | 14% | 39% (+25) | 39% | 43% (+4) | 35% | 41% (+6) |
| **Pass@8 Avg.** | | 51% | 55% (+4) | 32% | 65% (+33) | 56% | 65% (+9) | 59% | 64% (+5) |

*Table 2.* **After-GRPO** math results with models initialized by standard SFT v.s. vanilla PEAR-weighted NLL. The AVG. 64 rows report mean correctness over 64 sampled responses per question (a lower-variance estimate of per-sample Pass@1); single-sample Pass@1 can be noisy on small benchmarks, so averaging stabilizes the comparison. The **Pass@1 Avg.** row at the bottom averages these per-benchmark AVG. 64 values across benchmarks.

**Online RL** Starting from each offline checkpoint $\pi_0$, we run the same online RL procedure to obtain $\pi_{\text{RL}}$. We use GRPO (Shao et al., 2024) with learning rate $10^{-6}$, batch size 128, and KL coefficient 0.01.

### 4.3. Results

PEAR$_{B=1}$ stands for token-level weighting (§3.4), our default form of PEAR in the evaluation below that directly reflects the key intuition.

**PEAR improves post-RL performance under a fixed RL budget.** Figure 3a demonstrates the gain from initializing the model with the token-wise form of PEAR over standard SFT across different model sizes. We show a clear improvement on pass@1 across model sizes. Moreover,

PEAR outperforms all previously mentioned techniques in § 2. Notably, PEAR's performance does not surface in terms of its *out-of-the-box offline performance*, and even may not beat SFT, since PEAR is not designed to boost offline scores in isolation, but to better shape the prior for online RL.

In addition, we show in Table 2 that PEAR-initialized model can achieve higher overall performance across common math reasoning benchmarks for multiple model families.

Figure 3b shows that PEAR's weighting scheme can also work with KL-based knowledge distillation ($\ell_\theta = \text{KL}(\pi_\beta(\cdot \mid x, y_{<t}) \,\|\, \pi_\theta(\cdot \mid x, y_{<t}))$) and further improve upon that by computing the score using already-computed information during KD, adding minimal overhead to the KL-based KD baseline. PEAR$_{B=1}$ uses the exact same weighting as we apply it with NLL loss (i.e. suffix likelihood ratio).

| QWEN3-BASE | 0.6B | 1.7B | 4B | 8B |
|---|---|---|---|---|
| SFT +GRPO | 9% | 23% | 39% | 36% |
| SINGLE STEP +GRPO | 10% (+1) | 18% (-5) | 38% (-1) | 30% (-6) |
| PEAR_{B=1} +GRPO | 12% (+3) | 26% (+3) | 44% (+5) | 41% (+5) |

*Table 3.* Average accuracy across math benchmarks: () denote changes over SFT+GRPO (%). Negative gains shown in red.

| | DIRECT +GRPO | SFT +GRPO | PEAR_{B=1} +GRPO |
|---|---|---|---|
| QWEN3-0.6B-BASE | 2.8% | 8.4% | 13.1% (+4.7) |
| QWEN3-1.7B-BASE | 2.8% | 13.1% | 38.3% (+25.2) |
| QWEN3-4B-BASE | 8.0% | 49.5% | 59.8% (+10.3) |
| QWEN3-8B-BASE | 15.0% | 53.3% | 61.7% (+8.4) |

*Table 4.* PEAR can transfer to different RL task distribution.

This underscores PEAR 's generality as a plug-in reweighting approach for both commonly used offline objectives. Importantly, it proves our central hypothesis that offline stage should correct for distribution mismatch between behavior and target policies.

In Figure 8, we compare different modes of PEAR: Sequence-Level(§3.3), Token-Level(§3.4) and Block-Level(§3.5). All these variants out-perform standard SFT. Additionally, we observe that sequence-level weighing turns out highly effective despite its simplicity.

**You need to weigh the future, not a single action.** Concurrent works (Wu et al., 2025; Zhang et al., 2025a; Zhu et al., 2025c;a) propose several action-level stabilization to SFT of the form $\mathcal{L}(\theta) = \mathbb{E}_{(x,y,R)\sim\mathcal{D}} \left[ \sum_{t=1}^{T} \mathbf{w}(\mathbf{x}, \mathbf{y_{<t}}, \mathbf{y_t}) \ell_\theta(x, y_{<t}, y_t) \right]$, with $\mathbf{w}$ depending only on the prefix and the current action. We experiment with $w(x, y_{<t}, y_t) = \frac{\pi_\theta(y_t|x,y<t)}{\pi_\beta(y_t|x,y<t)}$ which is a generic form of one-step weighting, computed and stabilized the same way as $\Delta_t$ in PEAR.

This is a myopic objective that up-weights single actions that the target policy finds plausible, not taking into account the long-term effect. As shown in Figure 3a and Table 3, single-step weighting is less effective, since what matters for online RL readiness is whether the logged successful continuation is compatible with the current policy over the remaining horizon (Jiang & Li, 2016; Metelli et al., 2020; Nachum et al., 2019; Ross et al., 2011).We further compare PEAR against two concurrent importance-sampling-style baselines, DFT (Wu et al., 2025) and Proximal-SFT (Zhu et al., 2025c), on math reasoning (Figure 4) and SynLogic (Figure 5). PEAR dominates both across model families and benchmarks.

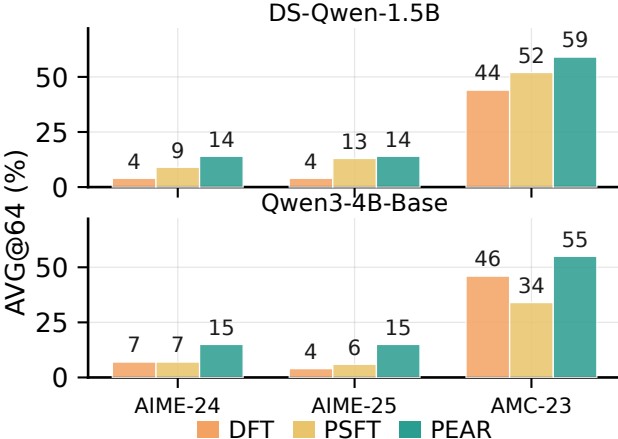

*Figure 4.* Comparison with importance-sampling baselines DFT (Wu et al., 2025) and Proximal-SFT (Zhu et al., 2025c) on math reasoning (AVG@64). PEAR consistently outperforms both on AIME-24, AIME-25, and AMC-23.

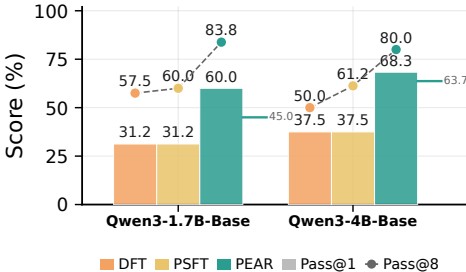

*Figure 5.* Baseline comparison on SynLogic. Bars = Pass@1 (best PEAR config), dots + dashed line = Pass@8. The small right-side tick on each PEAR bar marks PEAR's worst-config Pass@1 — even the worst PEAR run substantially beats both baselines.

### 4.4. PEAR Transfers to different RL task distributions.

We next test whether the capability induced by PEAR during offline training transfers to online RL on a different task distribution. Concretely, we initialize RL from the PEAR checkpoint and run online training on a subset of 12.8K problems from the Enigmata training set, then evaluate on a held-out set of Enigmata tasks after removing any near-duplicates across splits. As shown in Table 4, despite the difference between the offline training domain and the online RL domain, PEAR consistently provides a stronger initialization than standard SFT: it achieves better post-RL performance under the same RL recipe and roll-out budget. The benefit of PEAR is not overfit to the offline domain; it transfers better to a shifted online RL distribution under identical RL compute.

### 4.5. Black-Box PEAR: Proxy Behavior Policies

PEAR requires token log-probabilities from $\pi_\beta$ to compute importance weights. This is straightforward when the offline

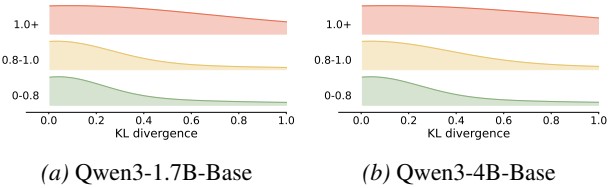

*(a)* Qwen3-1.7B-Base          *(b)* Qwen3-4B-Base

*Figure 6.* PEAR-to-base KL divergence across weight levels. $y$-axis is the weight (clipped). The token distribution is more heavily steered on important tokens that drive success probability.

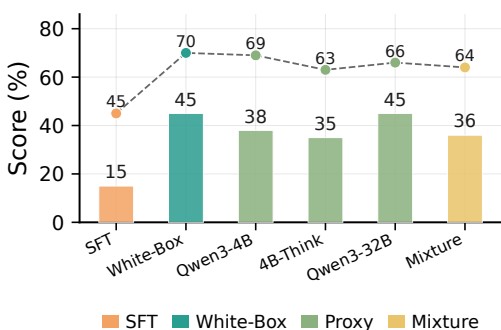

*Figure 7.* Black-box PEAR on SynLogic (Qwen3-1.7B-Base). Bars = Pass@1; markers = Pass@8. WHITE-BOX uses the true behavior policy $\pi_\beta$; remaining PEAR variants use an approximate proxy. All proxy variants substantially outperform SFT, and the strongest proxy (Qwen3-32B) matches or exceeds white-box. Full results including Qwen3-4B-Base are in App. A.

data is generated by a known open-weight model (as in our main experiments), but may be impractical when SFT data is curated or produced by a closed-source teacher. We show that PEAR remains effective when $\pi_\beta$ is approximated by a *proxy* model — a different Qwen3 variant not used to generate the data — or an *ensemble* of proxies.

As shown in Figure 7, every proxy variant substantially outperforms SFT+GRPO. The strongest proxy (Qwen3-32B) matches white-box PEAR on Pass@1 and comes within a few points on Pass@8. This demonstrates that PEAR does not necessarily require exact $\pi_\beta$ access in practice.

### 4.6. Incorporating Negative Examples

We experiment with the variant in §3.6, which pushes down the likelihood of entire negative sequences while avoiding token-/suffix-level signed ratio products that can be particularly unstable on long horizons. We sub-sample 50K positive data and included 50K negative data from identical behavior policy on the same set of instructions.

Figure 3a shows that under the same offline data budget, mixing negative trajectories for stabilization can bring significant additional gains to RL over positive-only PEAR initialization.

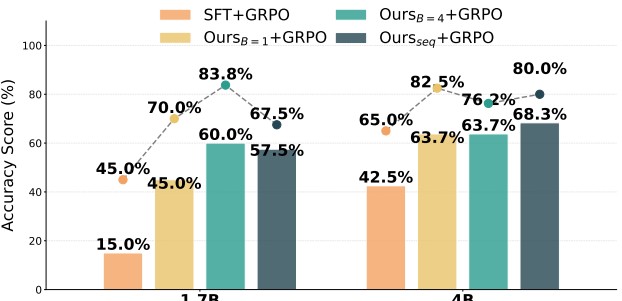

*Figure 8.* Performance of different variants of PEAR. The bars reflect Pass@1 and dots mark Pass@8.

### 4.7. Analysis

**RQ1: What Positions Does PEAR Concentrate On?** By design, PEAR's learning concentrates on the tokens that evaluate to larger weights. To analyze this effect, we compute per-token weight (§ 3.4) $\hat{G}_t$'s and $KL(\pi_\theta(a_t|s_t) \| \pi_\beta(a_t|s_t))$. As shown in Figure 6, the high-value tokens are distributionally more steered away from the base policy $\pi_\beta$, showing that the behavior of the trained model is systematically more updated on those important locations steering the suffix.

**RQ2: How Does PEAR's Learning Interact with Online RL?** Figure 10 shows the average principal angle between PEAR's gradients and those for GRPO is smaller than those of SFT and variants, suggesting that PEAR's correction can indeed make the offline updates more consistent with the online GRPO learning direction.

We also observe that applying stronger KL constraints could create greater mismatch between offline and online gradients, although it better preserves closeness to base model.

Thus, we observe online RL training after PEAR smallest drift measured by average NSS[1] in Figure 9-b between online and offline checkpoints compared with other initializations, whereas the heavy-lifting happened in the offline stage (Figure 9-a). It shows PEAR suffers the least from offline-to-online mismatch and spent less parameter updates correcting for those mis-alignments.

## 5. Related Works

**Learning Dynamics of Post-Training** There is a growing interest in understanding the learning characteristics of different post-training approaches (SFT v.s. RL).

A growing line of work studies why reinforcement-learning (RL) can behave qualitatively differently from supervised fine-tuning (SFT). They find that SFT more readily over-

---

[1]NSS measures the relative drift of a singular-value spectrum after training (Zhu et al., 2025b)

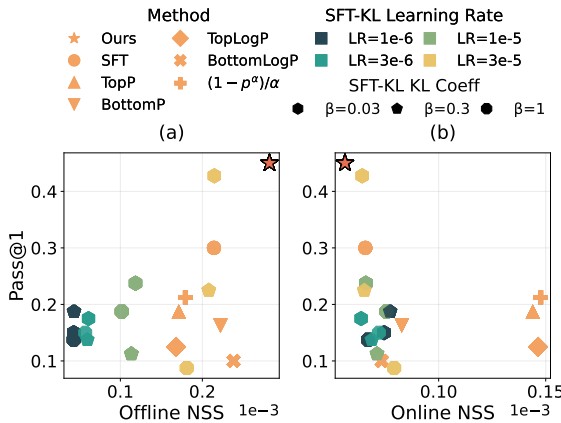

*Figure 9.* Parameter drift of different approaches. (a) is the NSS score between offline and base model. (b) is the NSS score between online and offline model.

fits and degrades out-of-distribution (OOD) performance, whereas on-policy RL more often improves generalization across distribution shifts and can partially undo SFT-induced drift (Chu et al., 2025; Jin et al., 2025b).

Recent analyses further connect RL's reduced catastrophic forgetting to its on-policy sampling bias,(Shenfeld et al., 2025; Chen et al., 2025a; Jin et al., 2025a). Beyond behavior-level metrics, recent analyses probe *parameter-space* dynamics: Zhu et al. (2025b) characterize RLVR updates as structured "off-principal" learning that preserves spectral structure relative to SFT, while Zhao et al. (2025) show RL post-training can amplify patterns already present in pretraining, often concentrating probability mass onto a dominant output mode.

**Offline RL For Language Models**   There is a line of work in LM post training that treat responses as logged decision trajectories (Lanchantin et al., 2025; Wang et al., 2024; Snell et al., 2023; Baheti et al., 2024; Richemond et al., 2024; Mukherjee et al., 2025a) and apply policy-optimization techniques to improve performance.

Others seek to improve online RL by introducing offline / semi-offline mechanisms (Lanchantin et al., 2025; Zhang et al., 2025b; Li et al., 2025b). Differently, our focus is on better bridging offline and offline stages in common SFT+RL post-training pipeline.

**Modifications To SFT**   Some works modify the SFT loss itself to reduce overfitting and capability loss—e.g., probability-based objectives beyond NLL (Li et al., 2025a), entropy-regularized distribution matching (Diao et al., 2026), and token/sample-wise reweighting or gating to suppress destructive gradients. (Sanyal et al., 2025; Lin et al., 2025).There are also various "importance-weighted / stabilized SFT" methods like iw-SFT (Qin & Springen-

berg, 2025), DFT (Wu et al., 2025), AFT (Zhu et al., 2025a), Proximal-SFT (Zhu et al., 2025c), OPC-SFT (Zhang et al., 2025a) that use probability-ratio or trust-region style weights primarily to mitigate off-policy instability, suppress the influence of low-probability tokens, and constrain KL/entropy drift so that supervised fine-tuning remains well-behaved under distribution shift. In contrast, our approach is not introduced as a stabilization or trust-region heuristic for SFT but a mechanism to better initialize models for subsequent online RL.

## 6. Conclusion

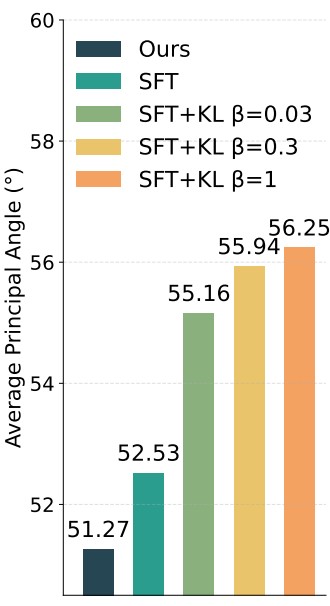

*Figure 10.* Mean principal angle between offline and online GRPO gradients.

Reasoning LLM post-training typically follows an offline SFT → online RL pipeline, so offline objectives should be judged by how well they initialize RL, not just by SFT accuracy. Across extensive experiments, we find that stronger offline performance is an unreliable proxy for post-RL performance: objectives that dominate after SFT can be overtaken after identical RL, producing substantial rank reversals.

We attribute this gap to offline-to-online policy mismatch. Offline SFT imitates logged continuations from logged prefixes, whereas online RL updates the model on trajectories sampled from its current policy, concentrating learning on prefixes it actually reaches. To reduce this mismatch, we propose PEAR (**P**olicy **E**valuation–inspired **A**lgorithm for Offline Learning Loss **R**eweighting), an OPE-inspired loss reweighting scheme that down-weights logged continuations that are implausible under the current policy and up-weights those that remain plausible. Empirically, PEAR consistently improves post-RL accuracy across verifiable reasoning games and math benchmarks, yielding up to 30 percentage points Pass@8 gain on AIME-2025 after online RL. More broadly, our results suggest a practical principle: the offline stage should prioritize *reproducible successes* under the target policy that will be optimized online.

## Acknowledgements

This work is supported by NSF Grant No. CHE2505932, an Amazon AICE Award, gift funding from AI2, and a grant from Coefficient Giving.

## Impact Statement

This paper presents work whose goal is to advance the field of Machine Learning. There are many potential societal consequences of our work, none of which we feel must be specifically highlighted here.

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

## Contents

## A. Black-Box PEAR: Proxy and Ensemble Behavior Policies

PEAR as presented in the main text assumes white-box access to the behavior policy $\pi_\beta$ so that token log-probabilities can be evaluated exactly. This holds whenever the offline data is generated by a known open-weight teacher (as in our main experiments) but can be too strong an assumption when SFT data is curated, scraped, or produced by a closed-source teacher. We show here that PEAR remains effective when $\pi_\beta$ is only available through an *approximate* proxy.

**Practical strategies.** We consider two ways to estimate $\pi_\beta$ without exact access: (i) a *single proxy* model from the same model family but with a different size or post-training (e.g. a different Qwen3 variant), and (ii) an *ensemble* of proxies, where we average per-token probabilities across multiple models.

**Setup.** We use the same SynLogic setting and SFT/RL recipe as in §4. The data is generated by Qwen3-8B (as in the main paper). We approximate $\pi_\beta$ using each of Qwen3-4B-Base, Qwen3-4B-Thinking, Qwen3-32B individually, and a uniform average over the three (Mixture), *excluding* the actual generator. We report Pass@1/Pass@8 after identical GRPO on Qwen3-1.7B-Base and Qwen3-4B-Base; for context we also report the white-box PEAR baseline.

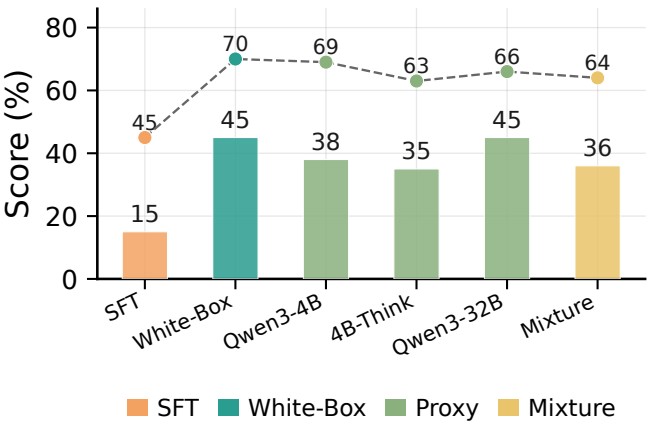

*Figure 11.* Black-box PEAR on SynLogic (Qwen3-1.7B-Base). Bars show Pass@1; dashed-line markers show Pass@8. SFT is the no-PEAR lower bound; WHITE-BOX uses the true behavior policy; remaining methods use an approximate $\pi_\beta$ (single-model proxies or the MIXTURE ensemble). Every proxy variant substantially beats SFT, and the strongest proxy (Qwen3-32B) matches white-box.

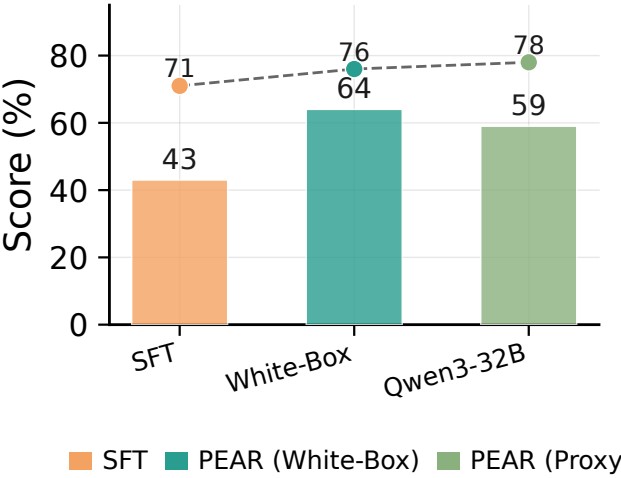

*Figure 12.* Black-box PEAR on SynLogic (Qwen3-4B-Base). PEAR with only a Qwen3-32B proxy still outperforms SFT on both metrics.

**Why does the proxy quality matter?** We compute the Pearson correlation between each proxy's per-token probabilities and those of the true behavior model across the dataset. Qwen3-4B-Thinking has the lowest correlation and correspondingly

the weakest post-RL performance; Qwen3-32B has the highest correlation and best proxy performance. This supports a simple practical heuristic: choose proxies whose token-level statistics align well with the data generator.

| Proxy | Pearson w.r.t. behavior |
|---|---|
| Qwen3-32B | 0.7328 |
| Qwen3-4B | 0.7237 |
| Ensemble (Mixture) | 0.7263 |
| Qwen3-4B-Thinking | 0.6605 |

*Table 5.* Pearson correlation between each proxy's per-token probabilities and the true behavior model.

**Takeaway.** Across all proxy variants, PEAR substantially outperforms SFT+GRPO. While exact $\pi_\beta$ remains preferred when available, these results show that PEAR can be operationalized in the more realistic setting where the data-generating policy is only approximately accessible.

## B. Hyperparameter Sensitivity: Clipping Range

PEAR uses log-importance clipping (§3.7) as a standard variance-reduction step. The clipping range is held fixed across all experiments in the paper; here we vary it to characterize sensitivity, on Qwen3-1.7B-Base trained on SynLogic.

We sweep $\min \in \{-0.08, -0.3\}$ and $\max \in \{0.08, 0.3\}$ on the per-decision log-ratio. Pre-RL Pass@32 averages 47.5% (std 3.95%) and Pass@64 averages 52.19% (std 2.58%). Variation across clip ranges is mild and does not lead to qualitative changes in behavior.

| Clip range | Pass@32 / Pass@64 |
|---|---|
| $[-0.08,\ 0.08]$ | 42.50 / 48.75 |
| $[-0.08,\ 0.30]$ | 46.25 / 52.50 |
| $[-0.30,\ 0.08]$ | 50.00 / 52.50 |
| $[-0.30,\ 0.30]$ | 51.25 / 55.00 |

*Table 6.* Sensitivity to clipping range on Qwen3-1.7B-Base, SynLogic. Differences are within roughly one standard deviation across settings.

## C. Block-Size Ablation

We ablate block size $B$ (§3.5) on Qwen3-1.7B-Base, SynLogic, comparing the three regimes: token-level ($B = 1$), block-level ($B \in \{4, 8\}$), and sequence-level (single global weight).

| Setting | Pass@1 | Pass@8 |
|---|---|---|
| Sequence | 57.50 | 67.50 |
| Block-4 | 60.00 | 83.75 |
| Block-8 | 68.75 | 77.50 |
| Block-1 (token) | 45.00 | 70.00 |

*Table 7.* Effect of granularity. Intermediate block sizes ($B \in \{4, 8\}$) balance fine-grained signal against multiplicative variance and outperform both extremes.

**Discussion.** The fully fine-grained end ($B = 1$) yields lower Pass@1, consistent with higher per-step variance in the multiplicative weight. The fully coarse end (Sequence) is also weakest on Pass@8: a single global weight applied to every token loses meaningful within-sequence variation. Intermediate $B$ provides a better stability–granularity trade-off, and we recommend it as a default.

## D. Beyond Reasoning: Instruction-Following

The main experiments target verifiable reasoning. To probe whether PEAR's benefit extends beyond math/logic, we apply the same SFT→RL pipeline to instruction-following.

**Setup.** We use the instruction-following split of SYNTHETIC-2 (Prime Intellect, 2025) for SFT, and AI2's Dolci-RL-Zero-IF-7B dataset for RL. The student model is Qwen3-0.6B-Base. We evaluate on IF-Eval and IF-Bench, two standard instruction-following benchmarks.

|  | SFT + GRPO | PEAR + GRPO |
|---|---|---|
| IF-Eval (Qwen3-0.6B-Base) | 37.34% | 57.67% |
| IF-Bench (Qwen3-0.6B-Base) | 16.67% | 29.59% |

*Table 8.* PEAR continues to outperform SFT+GRPO on instruction-following benchmarks by 20+ absolute points on IF-Eval, indicating the correction is not specific to math/logic.

**Discussion.** The gain (+20.3 pp on IF-Eval; +12.9 pp on IF-Bench) is comparable in magnitude to those observed on reasoning tasks. This is consistent with our central hypothesis: whenever the offline data generator differs from the model being optimized online, importance-weighted SFT yields a stronger initialization. Generalization to RLHF with *learned* reward models is a natural next step we leave for future work.

## E. Comparison with CHORD

CHORD (Zhang et al., 2025a) is a representative SFT/RL-hybrid method that mixes supervised and policy-gradient updates during RL. As acknowledged in Appendix D.1 of the CHORD paper, the method is not designed to cold-start from a base model: it operates at the RL stage, after the model is already a competent policy. PEAR operates at the SFT stage and replaces SFT before RL. We therefore view CHORD and PEAR as targeting different but complementary stages of the pipeline rather than competing methods. For empirical context, we run CHORD on the same SynLogic setting using Qwen3-0.6B-Base under matched SFT and RL data.

| Method | Pass@1 | Pass@8 |
|---|---|---|
| SFT | 10.00% | 40.00% |
| CHORD | 11.40% | 31.25% |
| PEAR | 12.50% | 42.50% |

*Table 9.* PEAR vs. CHORD on SynLogic with Qwen3-0.6B-Base. The two methods address different points of the SFT→RL pipeline; results are reported for completeness.

## F. Does PEAR Suppress Useful Low-Probability Trajectories?

A natural concern with any reweighting scheme is whether it inadvertently down-weights tokens whose "low" weight reflects genuine novelty rather than implausibility (e.g. formats, exploratory tokens). PEAR weights by *future plausibility* (suffix continuation), not by the current token's own likelihood, so the construction itself is self-correcting. We add three pieces of empirical evidence.

**(1) PEAR preserves reasoning patterns.** We count occurrences of common reasoning-pattern words ("wait", "hmm", "alternatively", "okay", "therefore") in generations from SFT- and PEAR-trained models on the same evaluation set. The counts are comparable; PEAR does not collapse reasoning patterns.

**(2) PEAR does not hurt learning of required answer formats.** Standard answer formats (e.g. \boxed{}) are reliably produced by PEAR-initialized models with no observable degradation in format compliance.

| Method | wait | hmm | alternatively | okay | therefore |
|--------|------|-----|---------------|------|-----------|
| SFT | 1691 | 76 | 234 | 176 | 300 |
| PEAR | 1540 | 74 | 267 | 158 | 244 |

*Table 10.* Reasoning-pattern token counts. PEAR maintains comparable usage of standard reasoning markers.

**(3) Cross-domain transfer.** PEAR's gains transfer across task distributions: e.g. offline training on SynLogic followed by RL on Enigmata still outperforms SFT initialization (Table 4 in the main paper). If PEAR suppressed useful structure, we would expect such transfer to break.

Together, these results indicate that PEAR's reweighting preserves rather than discards informative low-probability content.

## G. Compatibility with Other RL Algorithms (DAPO)

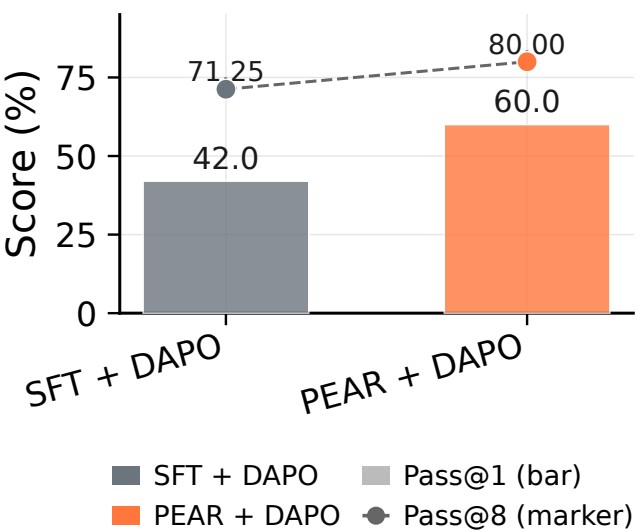

*Figure 13.* PEAR continues to outperform SFT when GRPO is replaced with DAPO (Yu et al., 2025), another widely used RLVR algorithm. Same Qwen3-1.7B-Base/SynLogic setup as §4.

To check that PEAR's benefit is not GRPO-specific, we replace GRPO with DAPO (Yu et al., 2025) and rerun the same Qwen3-1.7B-Base/SynLogic recipe used throughout §4. Figure 13 shows PEAR +DAPO outperforms SFT+DAPO by $+18$ Pass@1 and $+8.75$ Pass@8, mirroring the GRPO gains. The offline-to-online mismatch correction transfers across RLVR algorithms rather than being tied to GRPO.

## H. Computation Of Metrics

RLVR for LLMs has become increasingly expensive since the model needs to rollout on the training set and get updates at the same time. This gets even worse when the RLVR environment involves tool calling or code execution (Wei et al., 2025; Qian et al., 2025), which can take minutes to hours to finish. Therefore, understanding the signals that can suggest a model's potential after RLVR can save a huge amount of compute, and has received great attention in the LLM community (Sun et al., 2026). In this section, we discuss the implementation details of the signal metrics we evaluated.

### H.1. KL

We compute the forward KL between the base model and the model trained on offline data to understand how much the trained model's distribution diverge from the original one, and whether big or small divergence result in superior performance. Specifically, the metric we evaluated is defined as follows:

$$\text{Forward KL} = \text{KL}(P_{\text{base}}||P_{\text{trained}}) = \Sigma P_{\text{base}} \log\left(P_{\text{base}}/P_{\text{trained}}\right)$$

To evaluate the forward KL, we first collect a calibration set of 269 question-answer pairs in SynLogic games, and forward the data through the base and tuned model to compute the distribution and corresponding KL. The reported forward KL is computed by taking macro average over all sequences in the calibration set.

## H.2. Sparsity

Mukherjee et al. (2025b); Zhu et al. (2025b) observed the different sparsity patterns in SFT and RLVR training, we try to understand if update patterns will have different sparsities under different offline learning objectives. Given a linear module $W_{\text{base}}$ from the base model and the corresponding module $W_{\text{trained}}$ in the trained model, $\epsilon$ be the sparsity threshold, the sparsity of the module is calculated as:

$$\Delta W = W_{\text{trained}} - W_{\text{base}},$$
$$\text{sparsity}(W) = \frac{1}{|\Delta W|} \sum_{i,j} \mathbf{1}(|\Delta W_{ij}| < \epsilon),$$

A model's sparsity is calculated by taking a macro average across all its linear modules.

## H.3. Normalized Spectrum Shift

Normalized Spectrum Shift (NSS) (Zhu et al., 2025b) is a metric to measure the drift of the tuned model in the parameter space. For each module, we first perform singular value decomposition on the weight matrices to obtain the singular values, then measure the normalized distance between the singular value spaces.

$$\text{NSS}(W) = \|\sigma(W_+) - \sigma(W_0)\|_2 / \|\sigma(W_0)\|_2$$

## H.4. Gradient Rotation

We compute the rotation of gradients during the offline and online stage to understand if the offline and online training stages update the model in similar directions. Specifically, given the gradient if a module in the base model $\nabla_{\text{offline}} W$ in the offline stage and the gradient of the same module in the online stage $\nabla_{\text{online}} W$, we first perform SVD on the two matrices to obtain $U_{\text{offline}}$ and $U_{\text{online}}$, then we evaluate the subspace rotation between the two matrices as follows:

$$\cos_{\theta_i}(U) := \sigma_i(U_{\text{offline},k}^\top U_{\text{online},k}), i = 1, \ldots, k$$

where $U_K$ denotes and top-k subspace of $U$ and $k$ is equal to 128 in our case. To estimate the rotation of a module we simply take the average of the $\theta_i, \ldots, \theta_k$. To estimate the rotation of model, we simply take macro average over all the linear modules.

In order to compute the gradients, we take a model that has been trained on offline data, and use another calibration set to run a pilot offline training on the model for 10 steps and collect the gradients. For the online gradient, we simply run GRPO on the model for 10 steps and collect the gradients. We take the means of the offline and online gradients across the steps and use the means to compute the module-wise rotations.

## H.5. Results

The results of metrics vs online pass@1 for Qwen3-1.7B-Base are visualized in Figure 14. From the figure, we can see that IS-SFT results in comparable forward KL divergence and much lower sparsity compared to SFT methods. Additionally, our method results in greater update during the offline stage, causing greater spectrum drift against the base model compared to SFT methods. In contrast, IS-SFT causes smaller updates during the online stage.

# I. Suffix Change-of-Measure

Let $x$ be a prompt and $y_{1:T}$ a token trajectory. For any $t \in \{1, \ldots, T\}$, define the prefix $s_t := (x, y_{<t})$ and the suffix $y_{t:T}$. For an autoregressive policy $\pi$,

$$\pi(y_{t:T} \mid s_t) = \prod_{k=t}^{T} \pi(y_k \mid x, y_{<k}).$$

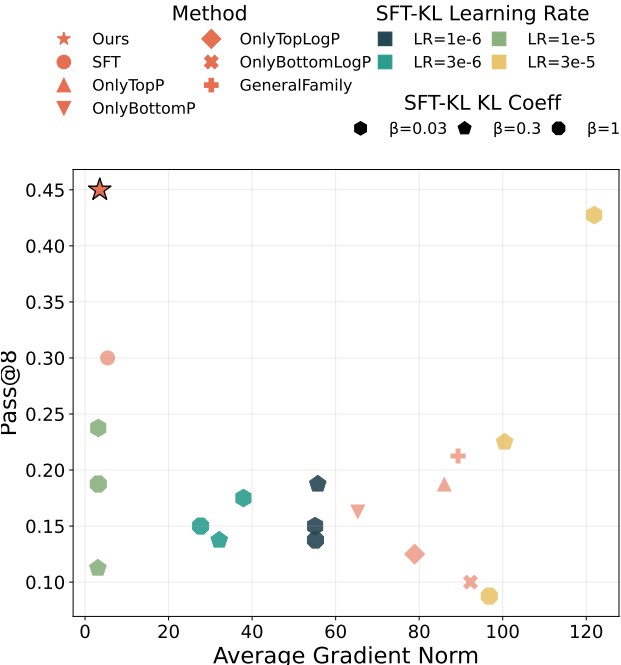

*Figure 14.* The comparison between different metrics versus SynLogic online pass@1. (a) offline model forward KL divergence against the base model. (b) offline model update sparsity against the base model. (c) average spectrum drift of different linear modules in the base and offline model. (d) average spectrum drift of different linear modules in the offline and online model.

Define the *suffix likelihood ratio*

$$\rho_{t:T}(x, y) := \frac{\pi_\theta(y_{t:T} \mid s_t)}{\pi_\beta(y_{t:T} \mid s_t)} = \prod_{k=t}^{T} \frac{\pi_\theta(y_k \mid x, y_{<k})}{\pi_\beta(y_k \mid x, y_{<k})}.$$

Then for any measurable function $\varphi$ of the continuation (and the fixed prefix),

$$\mathbb{E}_{y_{t:T} \sim \pi_\theta(\cdot \mid s_t)}\big[\varphi(s_t, y_{t:T})\big] = \sum_{y_{t:T}} \pi_\theta(y_{t:T} \mid s_t)\, \varphi(s_t, y_{t:T})$$

$$= \sum_{y_{t:T}} \pi_\beta(y_{t:T} \mid s_t)\, \frac{\pi_\theta(y_{t:T} \mid s_t)}{\pi_\beta(y_{t:T} \mid s_t)}\, \varphi(s_t, y_{t:T})$$

$$= \mathbb{E}_{y_{t:T} \sim \pi_\beta(\cdot \mid s_t)}\big[\rho_{t:T}(x, y)\, \varphi(s_t, y_{t:T})\big],$$

assuming $\pi_\beta(y_{t:T} \mid s_t) > 0$ whenever $\pi_\theta(y_{t:T} \mid s_t) > 0$.

*Intuition.* Conditioning on the same prefix $s_t$, the two policies induce different distributions over the remaining continuation $y_{t:T}$. The ratio $\rho_{t:T}$ is exactly the Radon–Nikodym derivative that reweights $\pi_\beta$-suffix samples into unbiased expectations under $\pi_\theta$: suffixes that are more likely under $\pi_\theta$ than $\pi_\beta$ receive larger weight, and vice versa.

## J. An Alternative Intuition: Suffix ratios as an off-policy estimate of return / value.

Recall the (discounted) terminal-feedback action-value under the *target* policy:

$$Q_\gamma^{\pi_\theta}(s_t, a_t) \triangleq \mathbb{E}_{a_{t+1:T} \sim \pi_\theta(\cdot \mid s_{t+1:T})}\big[\gamma^{T-t}\, R(\tau) \mid s_t, a_t\big], \tag{1}$$

where $\tau = (s_1, a_1, \ldots, s_T, a_T)$ and $R(\tau)$ is observed only at $T$.

*Change of measure on the continuation.* Condition on the same prefix decision $(s_t, a_t)$ and rewrite the $\pi_\theta$-continuation expectation as an expectation over continuations sampled from the logging policy $\pi_\beta$:

$$
\begin{aligned}
Q_\gamma^{\pi_\theta}(s_t, a_t) &= \sum_{a_{t+1:T}} \pi_\theta(a_{t+1:T} \mid s_t, a_t) \, \gamma^{T-t} R(\tau) \\
&= \sum_{a_{t+1:T}} \pi_\beta(a_{t+1:T} \mid s_t, a_t) \, \underbrace{\frac{\pi_\theta(a_{t+1:T} \mid s_t, a_t)}{\pi_\beta(a_{t+1:T} \mid s_t, a_t)}}_{w_{t+1:T}(\tau)} \, \gamma^{T-t} R(\tau) \\
&= \mathbb{E}_{\tau \sim \pi_\beta} \left[ \gamma^{T-t} R(\tau) \, w_{t+1:T}(\tau) \mid s_t, a_t \right],
\end{aligned}
\tag{2}
$$

with the suffix importance ratio

$$
w_{t+1:T}(\tau) \triangleq \prod_{j=t+1}^{T} \frac{\pi_\theta(a_j \mid s_j)}{\pi_\beta(a_j \mid s_j)}.
$$

*Intuition:* $w_{t+1:T}$ "translates" logged suffixes into what $\pi_\theta$ would typically see. If the logged continuation is unlikely under $\pi_\theta$, it should contribute little to $\pi_\theta$'s expected return-to-go from $(s_t, a_t)$.

**From $Q$ to a token-level return estimator.** Given one logged trajectory $\tau \sim \mathcal{D}$, a single-sample plug-in estimator of (2) is exactly your per-token credit weight

$$
G_t(\tau) \triangleq \gamma^{T-t} R(\tau) \, w_{t+1:T}(\tau), \qquad \text{so that} \quad G_t(\tau) \approx Q_\gamma^{\pi_\theta}(s_t, a_t).
\tag{3}
$$

*Intuition:* uniform SFT corresponds to replacing $Q$ by a constant (every token gets equal credit), while PEAR replaces it by an outcome-aware return estimate that (i) propagates terminal feedback back to earlier decisions (via $\gamma^{T-t} R$) and (ii) discounts suffixes that $\pi_\theta$ would not actually realize during on-policy rollouts (via $w_{t+1:T}$).

# K. Details On Baselines

## K.1. Token-Adaptive Loss Reweighting (TALR)

TALR reweights token-level negative log-likelihood (NLL) by an exponential function of token difficulty. Given token probability $p_t$ for the supervised token at position $t$, define token NLL

$$
\ell_t = -\log p_t.
$$

TALR assigns an adaptive weight

$$
\tilde{w}_t = \exp\left(-\frac{\ell_t}{\tau}\right), \qquad w_t = \max\big(\text{sg}(\tilde{w}_t), w_{\min}\big),
$$

where $\text{sg}(\cdot)$ denotes *stop-gradient* (weights treated as constants in backprop).

The reweighted batch loss is the (token-)mean:

$$
\mathcal{L}_{\text{TALR}} = \frac{1}{N} \sum_{t=1}^{N} w_t \, (-\log p_t),
$$

with $N$ the number of supervised tokens in the batch.

**Key TALR hyperparameters.**

- **Weight floor** $w_{\min}$: fixed to 0.01 in all experiments (prevents vanishing weights on very hard tokens).

- **Temperature** $\tau$: selected **dynamically** each step as the **median of the average sequence loss within the batch**.

| Key | Name | Per-token loss $f(p)$ | Hyperparameters / mask |
|---|---|---|---|
| `original` | NLL (standard SFT) | $-\log p$ | None |
| `GeneralFamily-`$\alpha$ | Probability family | $\dfrac{1-p^\alpha}{\alpha}$ | $\alpha$ (with $\alpha \to 0$ recovering $-\log p$) |
| `p` | Plain-$p$ objective | $1-p$ | None (equiv. to `GeneralFamily-1` up to constants) |
| `OnlyTopP-`$q$ | Top-thresholded (plain-$p$) | $(1-p)\,\mathbf{1}[p \geq q]$ | $q \in [0,1]$ |
| `OnlyBottomP-`$q$ | Bottom-thresholded (plain-$p$) | $(1-p)\,\mathbf{1}[p \leq q]$ | $q \in [0,1]$ |
| `OnlyTopLogP-`$q$ | Top-thresholded (NLL) | $-\log(p)\,\mathbf{1}[p \geq q]$ | $q \in [0,1]$ |
| `OnlyBottomLogP-`$q$ | Bottom-thresholded (NLL) | $-\log(p)\,\mathbf{1}[p \leq q]$ | $q \in [0,1]$ |

*Paper-only (used for analysis/ablations; not exposed as repo keys)*

*Table 11.* Objectives in *Beyond Log Likelihood*. Here $p := p_\theta(y_t \mid y_{<t}, x)$ denotes the model probability of the ground-truth token at step $t$, and the sequence loss is $\sum_t f(p_t)$.

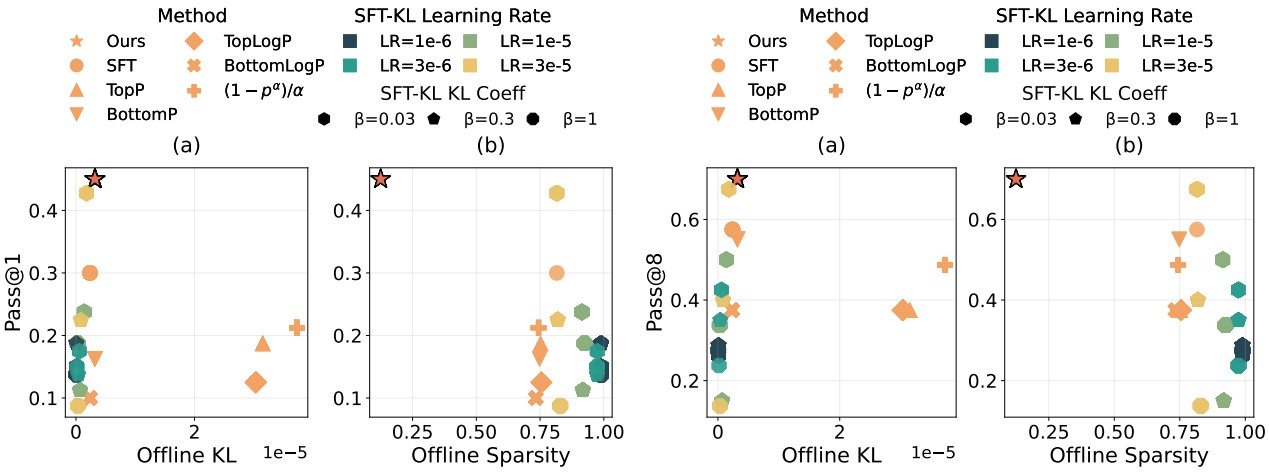

*(a)* KL-to-base and update sparsity of offline updates versus Pass@1. *(b)* KL-to-base and update sparsity of offline updates versus Pass@8.

*Figure 15.* Offline update strength measured by KL to base model and sparsity of parameter updates.

### K.2. Beyond Log Likelihood

Here we summarize the hyper-parameter settings reported in Li et al. (2025a).

The base learning rate is $5 \times 10^{-5}$.

**Objective hyperparameters.** Their core parametric family is $f_\alpha(p) = \frac{1-p^\alpha}{\alpha}$ (with $\alpha \to 0$ recovering NLL). In the main math results, they instantiate several concrete choices, including: (i) NLL $-\log p$, (ii) $-p$ (equivalently the $\alpha = 1$ member up to an additive constant), and (iii) a hard-thresholded NLL of the form $-\log(p) \cdot \mathbb{I}[p \geq 0.2]$. They also discuss higher-power prior-leaning variants (e.g., $\alpha = 10$).

## L. Discussion: Should Offline Training Match Online RL Characteristics In The Two-Stage Process?

While it is tempting to treat common stabilization signals—smaller KL to the base (Shenfeld et al., 2025) policy, sparser (Mukherjee et al., 2025b) (lower-magnitude) updates, or a smaller "rotation" (Zhu et al., 2025b) away from the base representation, these quantities primarily measure *conservatism*, not actually an accurate "mismatch correction".

Figure 9-a shows that KL penalties ensure smaller drift yet does not boost the performance compared with vanilla SFT, and while PEAR leads to more aggressive KL drifts and denser updates 9-b, it performs better. In fact, for downstream RL the goal is not to minimize movement per se, but to move in the *right directions*: toward behaviors that improve expected return under on-policy rollouts, even if that requires nontrivial deviation from the base. Consequently, a checkpoint that "looks

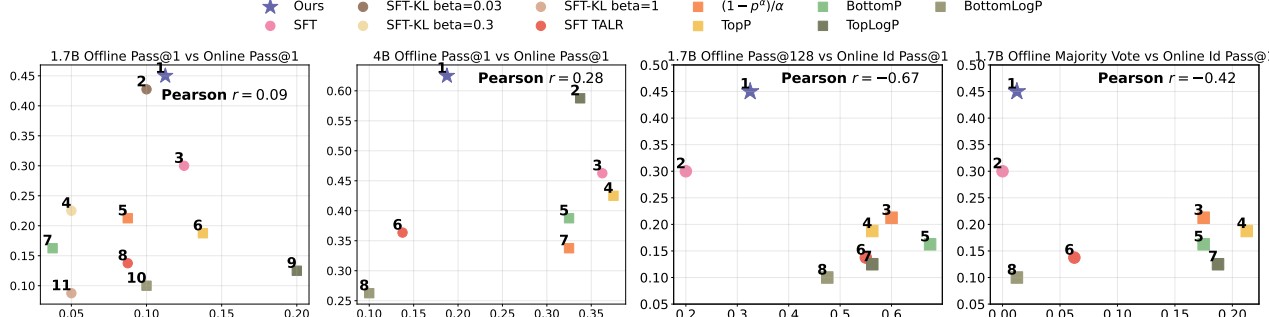

*Figure 16.* Visualization of offline vs online performance. (**a**): Qwen3-1.7B-Base-Base offline pass@1 versus online pass@1. (**b**): Qwen3-4B-Base offline pass@1 versus online pass@1. (**c**): Qwen3-1.7B-Base offline pass@128 versus online pass@1. (**d**): Qwen3-1.7B-Base offline majority vote versus online pass@1.

stable" by these proxies may still yield weak post-RL gains (or learn slowly) because it has not acquired the right inductive biases, coverage, or credit-assignment structure that makes subsequent RL compute-efficient.

## M. Table For Offline vs Online Metrics (Pass@1 and Pass@8)

One surprising finding we made is that stronger offline performance does not necessarily lead to stronger post-RL performance. The results are visualized in Figure 16. This section shows the detailed offline and online performances of Qwen3-1.7B-Base and Qwen3-4B-Base trained with different learning objectives. From the figures we can clearly see the line segments intersect with each other, and that offline-online performances do not have a consistent ranking. Detailed evaluation results of Qwen3-1.7B-Base and Qwen3-4B-Base can be found in Table 12 and 13, respectively.

| Objective | Offline | | | | Online | |
|---|---|---|---|---|---|---|
| | Pass@1 | Pass@8 | Pass@128 | maj vote | Pass@1 | Pass@8 |
| PEAR | 32.50% | 37.50% | 31.25% | 1.25% | 22.50% | 56.25% |
| PEAR | 20.00% | 23.75% | 32.50% | 1.25% | 45.00% | 70.00% |
| SFT | 22.50% | 27.50% | 20.00% | 0.00% | 30.00% | 57.50% |
| SFT TALR | 8.75% | 35.00% | 55.00% | 6.25% | 13.75% | 48.75% |
| SFT-KL($\beta = 0.03$) | 10.00% | 35.00% | - | - | 42.75% | 67.50% |
| SFT-KL($\beta = 0.3$) | 5.00% | 40.00% | - | - | 22.50% | 40.00% |
| SFT-KL($\beta = 1.0$) | 5.00% | 28.75% | - | - | 8.75% | 13.75% |
| TopP | 13.75% | 33.75% | 56.25% | 21.25% | 18.75% | 37.50% |
| TopLogP | 20.00% | 37.50% | 56.25% | 18.75% | 10.00% | 37.50% |
| BottomP | 3.75% | 30.00% | 56.25% | 13.75% | 16.25% | 55.00% |
| BottomLogP | 10.00% | 28.75% | 47.50% | 1.25% | 10.00% | 37.50% |
| $(1-p)^\alpha/\alpha$ | 8.75% | 27.50% | 60.00% | 17.50% | 21.25% | 48.75% |

*Table 12.* Offline and online pass rates of Qwen3-1.7B-Base with different learning objectives. All results are evaluated on SynLogic.

| Objective | Offline | | Online | |
|---|---|---|---|---|
| | Pass@1 | Pass@8 | Pass@1 | Pass@8 |
| PEAR | 32.50% | 37.50% | 63.75% | 76.25% |
| SFT | 10.00% | 26.25% | 42.50% | 71.25% |
| SFT TALR | 13.75% | 45.00% | 36.40% | 68.80% |
| TopP | 37.50% | 45.00% | 42.50% | 63.75% |
| TopLogP | 33.75% | 52.50% | 58.76% | 72.50% |
| BottomP | 32.50% | 60.00% | 38.75% | 71.25% |
| BottomLogP | 10.00% | 16.25% | 26.25% | 42.50% |
| $(1-p)^{\alpha}/\alpha$ | 32.50% | 47.50% | 33.75% | 63.75% |

*Table 13.* Offline and online pass rates of Qwen3-4B-Base with different learning objectives. All results are evaluated on SynLogic.

