# OpenReview forum: "Good SFT Optimizes for SFT, Better SFT Prepares for Reinforcement Learning"
_ICML.cc/2026/Conference — ICML 2026 regular_

### Official Review · Reviewer_MmtE · 2026-03-10

**Soundness:** 3
**Presentation:** 3
**Significance:** 2
**Originality:** 2
**Overall Recommendation:** 4
**Confidence:** 5

**Summary:**

This paper investigates the post-training pipeline(SFT/RL) of LLMs. The authors identify a critical issue: optimizing SFT for offline performance can result in sub-optimal initialization for the subsequent RL stage, leading to poorer final performance. This degradation is attributed to a distribution mismatch between the offline SFT data and the trajectories explored during online RL. To mitigate this, the authors propose PEAR. PEAR uses importance sampling to reweight the SFT loss at the token, block, or sequence level, down-weighting offline continuations that are unlikely under the current model policy.

**Compliance With Llm Reviewing Policy:**

Affirmed.

**Final Justification:**

Thanks for the author's reply. Some doubts have been clarified, and I will raise my score to 4 points.

**Key Questions For Authors:**

1. Table 2 shows the quantitative indicators, and we found that the data for ds-qwen-1.5b is much higher than that for other models. What could be the reason for this? In addition, pass@1 is not included in this table.
2. Include at least one experiment evaluating PEAR's effectiveness on a standard open-ended alignment benchmark. I'm also curious to compare PEAR with the method of RL-SFT alignment in the loss.

**Limitations:**

yes

**Strengths And Weaknesses:**

Strengths:
The paper excellently identifies and articulates the distribution mismatch between offline behavior policies and online target policies in LLM post-training. PEAR is introduced as a straightforward, plug-and-play reweighting scheme.

Weaknesses:
1. Insufficient experimental scope: This empirical evaluation focuses only on verifiable reasoning domains (mathematical and logical puzzles). The performance of PEAR in general scope has not been included in this paper. It only shows the performance of the accuracy metric, and the analysis of why the method is advantageous is insufficient.
2. Lack of comparative analysis. There are already many articles discussing IS between RL and SFT. These articles only compare direct SFT+RL and SFT+PEAR, and lack comparative experiments with these improved methods.
3. Lack of mathematical proof. This is a theoretical problem, but no mathematical proof of the method has been found.
4. Writing errors. pass@128 in figure1; 'In addition' not 'In addtion' in Figure 1 caption; 'by by' in line 95;

---

> ### Author Rebuttal · Authors · 2026-03-31
>
> **TLDR**:
> - **Scope**: We present new results to show PEAR works beyond the reasoning domain by new experiments on instruction following.
> - **Baselines**: We now additionally include DFT and PSFT on top of the 10+ baselines already in the paper, and PEAR still performs best after RL. This further supports our central claim that preparing for RL requires accounting for the future continuation, rather than only the current token.
> - **Other clarifications** please see below!
>
> ---
>
> > W1: Applicability beyond reasoning
>
> Although PEAR is primarily motivated in reasoning scenario,our new instruction-following results show the idea is not limited to math/logic puzzles.
> We perform SFT/PEAR on the instruction-following split of SYNTHETIC-2, and RL using allenai’s Dolci-RL-Zero-IF-7B dataset. We use Qwen3-0.6B-Base as the student model and show PEAR’s advantage on two commonly-used instruction-following benchmarks: IF-Eval and IF-Bench. This shows PEAR’s effectiveness on domains beyond reasoning and can be extended to broader domains like instruction following.
>
> | Model           | IF-Eval (SFT + GRPO) | IF-Eval (PEAR + GRPO) | IF-Bench (SFT + GRPO) | IF-Bench (PEAR + GRPO) |
> | --------------- | -------------------: | --------------------: | --------------------: | ---------------------: |
> | Qwen3-0.6B-Base |               37.34% |                57.67% |                16.67% |                 29.59% |
>
>
> ---
>
> > W2: Additional Baselines
>
> In our paper, we experimented with basic form single-step importance-sampling correction (Figure 3, Single Step IS + GRPO) in addition to baselines in Figure 1 (10+ in total).
>
> Here, we compare with two new IS baselines (both cited in the paper): DFT and PSFT(Proximal-SFT) on math reasoning benchmarks (same setting as in the paper).
> DFT re-weights each token with a single-step importance ratio (with constant behavior approximation) and PSFT adds a PPO-style trust-region / clipping constraint.
> The table below shows post-GRPO average accuracy across 64 samples per question.
>
> PEAR + GRPO yields significantly better performance. As discussed in Sec 4.3, one needs to weigh the future, not the current token, to better prepare for RL.
>
> | Benchmark | Method | DS-Qwen-1.5B | Qwen3-4B-Base |
> | --------- | -----: | -----------: | ------------: |
> | AIME24    |    DFT |            4 |             7 |
> | AIME24    |   PSFT |            9 |             7 |
> | AIME24    |   PEAR |           14 |            15 |
> | AIME25    |    DFT |            4 |             4 |
> | AIME25    |   PSFT |           13 |             6 |
> | AIME25    |   PEAR |           14 |            15 |
> | AMC23     |    DFT |           44 |            46 |
> | AMC23     |   PSFT |           52 |            34 |
> | AMC23     |   PEAR |           59 |            55 |
>
> ---
>
> > W3: “This is a theoretical problem, but I did not find mathematical proof.”
>
> Our contribution is methodological and empirical: we introduce a practical training objective for improving the offline-to-online transition and support it through extensive controlled experiments (roughly 80 offline algo -> online RL experiments) and mechanistic analysis.
> That said, we included the mathematical derivation and the value-based intuition in appendices B and C, and are happy to discuss the mathematical properties the reviewer would suggest. We will also clarify this positioning of our work in the next revision.
>
> ---
>
> > W4: Clarification on Pass@1 Metric
>
> In the table, the AVG.64 entries are the average correctness over 64 sampled responses per question, i.e., a lower-variance estimate of per-sample accuracy / pass@1. We used this estimate because single-sample pass@1 can be noisy, and averaging over 64 samples yields a more stable comparison. We explained that in the caption and will make it clearer in the next revision!
>
> ---
>
> > Why DeepSeek Distilled model sees the most significant improvement.
>
> We thank the reviewer for this great observation! DS-Qwen-1.5B is a product of cross-family distillation. This creates a larger offline-to-online mismatch — precisely the condition PEAR is designed to correct. The larger gains are therefore consistent with our central hypothesis: the greater the behavior–target divergence, the more PEAR's importance-weighted correction contributes.
>
> ---
> > Clarifying Figure Captions
>
> We want to clarify that the third subplot aims to show pass@K (K is large) also does not indicate online pass@1 performance. I will make this clear in the next revision, and fix other typo & improve formatting!
>
> ---
> > Analysis on why PEAR works
>
> We included mechanism-oriented analyses beyond final accuracy in the paper.  First, the analysis shows PEAR concentrates updates on high-weight positions. Second, PEAR’s offline gradients are more aligned with GRPO gradients. Third, PEAR-initialized models exhibit less corrective parameter drift during RL. We will surface these analyses more prominently in the main text to better support the central mismatch hypothesis.

---

> > ### Author Rebuttal · Reviewer_MmtE · 2026-04-04
> >
> > Thanks for the author's reply. Some doubts have been clarified, and I will raise my score to 4 points.

---

> > > ### Author Response · Authors · 2026-04-04
> > >
> > > Thank you for your response — we are glad to hear that our rebuttal helped address your concerns, and we sincerely appreciate your re-evaluation of our work. We would be very happy to continue the discussion if there are any remaining questions or aspects that we could further clarify to strengthen the paper!

---

### Official Review · Reviewer_Bcfh · 2026-03-12

**Soundness:** 3
**Presentation:** 3
**Significance:** 3
**Originality:** 3
**Overall Recommendation:** 4
**Confidence:** 4

**Summary:**

This paper studies the widely used SFT-RL pipeline for reasoning LLMs and shows that stronger SFT checkpoints do not necessarily yield better performance after identical downstream RL. To address the offline-to-online distribution mismatch between a behavior policy and the target policy, the authors propose PEAR, an off-policy-evaluation–inspired reweighting scheme applied during SFT. PEAR computes importance ratios between the target and behavior policies and uses them to reweight token losses at sequence, block, or token-suffix levels with stabilization (discounting, clipping). Experiments on synthetic logic games and math reasoning across multiple model families indicate consistent post-RL gains, including notable improvements on AIME-2025.

**Compliance With Llm Reviewing Policy:**

Affirmed.

**Key Questions For Authors:**

1. There are some formatting issues in the paper.

**Limitations:**

yes

**Strengths And Weaknesses:**

**Strengths**

1. The paper tackles the highly prevalent and critical SFT-to-RL post-training pipeline in modern LLMs.
2. The proposed method is conceptually intuitive, elegantly motivated, and backed by solid mathematical and empirical evidence.
3. The empirical finding that "higher offline SFT accuracy does not guarantee a better RL initialization" is highly thought-provoking for the community.

**Weaknesses**

1. The method relies on highly specific hyperparameters (e.g., clipping bounds). It is concerning whether PEAR requires exhaustive, dataset-specific tuning. Its algorithmic robustness needs further justification.
2. Experiments are strictly confined to Math and Logic domains using rule-based RLVR. The method’s effectiveness on general reasoning tasks and standard RLHF (with learned reward models) remains unverified.
3. The evaluation lacks comparisons with recent SFT/RL hybrid methods that also aim to improve the SFT-to-RL transition, such as CHORD [1].
4. By down-weighting "implausible" offline trajectories, might PEAR hinder the model's ability to acquire entirely novel knowledge or adapt to specific response formats during SFT, ultimately degrading the final SFT+RL performance?

[1] On-Policy RL Meets Off-Policy Experts: Harmonizing Supervised Fine-Tuning and Reinforcement Learning via Dynamic Weighting.

---

> ### Author Rebuttal · Authors · 2026-03-31
>
> **TL;DR:**
>
>
> **(1) Hyperparameters:** A single shared clip configuration is used across all experiments; sensitivity analysis on Qwen3-1.7B-Base confirms stable performance (Pass@32 std 3.95%, Pass@64 std 2.58%) across clip ranges.
>
>
> **(2) Scope:** New instruction-following results show PEAR+GRPO outperforms SFT+GRPO by 20+ points on IF-Eval (37.3% -> 57.7%), extending beyond math/logic.
>
>
> **(3) Baselines:**
> - Result and discussion on CHORD included.
> - DFT and Proximal-SFT added; PEAR dominates post-GRPO across all models.
>
> **(4) Knowledge suppression:** We explain PEAR does not weigh based on token difficulty but future plausibility. We list 3 points (with new analysis) to support.
>
>
> **(5) Format:** We will improve formatting in the next revision!
>
>
> ---
>
>
> > **W1: Hyperparameter Analysis**
>
>
> We clarify that PEAR’s clipping terms are standard stabilization, not dataset-specific hyperparameters. We used the same configuration across models and tasks throughout the paper, and will make this explicit in the revision.
>
>
> Below is hyperparameter analysis with clip min and clip max on per-token ratios on Qwen3-1.7B-Base. Clip range for log importance ratio: Min {-0.08, -0.3} × max {0.08, 0.3}. Pass@32 47.5% std 3.95%, pass@64 52.19% std 2.58% (pre-RL). We will include the table in the next revision. The hyperparameter variation indeed causes mild fluctuations but does not cause drastic behavior changes.
>
>
>
>
> | Clip range   | pass@32 / pass@64 |
> |--------------|-------------------|
> | [-0.08,0.08] | 42.50 / 48.75     |
> | [-0.08,0.3]  | 46.25 / 52.50     |
> | [-0.3,0.08]  | 50.00 / 52.50     |
> | [-0.3,0.3]   | 51.25 / 55.00     |
>
>
> ---
>
>
> > **W2: Applicability Beyond Math and Logic**
>
>
> We agree a broader applicability would strengthen the approach!
> PEAR is inspired by and designed for reasoning-centric post-training which is inherently search-like and step-wise. We add experiments on instruction-following.
>
>
> To test whether the idea extends beyond pure math and logic puzzles, we perform SFT/PEAR on the instruction-following split of SYNTHETIC-2, and RL using AI2’s Dolci-RL-Zero-IF-7B dataset. We use Qwen3-0.6B-Base as the student model and show PEAR’s advantage on two commonly-used instruction-following benchmarks: IF-Eval and IF-Bench. This proves PEAR’s effectiveness on domains beyond logic puzzles and math.
>
>
> | Model             | IF-Eval: SFT + GRPO | IF-Eval: PEAR + GRPO | IF-Bench: SFT + GRPO | IF-Bench: PEAR + GRPO |
> |------------------|---------------------:|----------------------:|----------------------:|-----------------------:|
> | Qwen3-0.6B-Base  | 37.34%               | 57.67%                | 16.67%                | 29.59%                 |
>
>
> ---
>
>
> > **W3: Additional Baselines and Related Approaches**
>
> CHORD and other RL-with-supervision works present another promising alternative to post-training. We ran CHORD with the same SFT & RL data on SynLogic using Qwen3-0.6B-Base. Yet this does not fully reflect CHORD’s intended regime: *Appendix D.1* of CHORD highlights that the algorithm is not designed to cold-start a base model; instead, it operates at a different stage than PEAR, which replaces SFT from the base model.  Thus we see CHORD and other hybrid approaches  as complementary to PEAR rather than competing.
> | Method | Pass@1 | Pass@8 |
> | ------ | -----: | -----: |
> | SFT | 10.00% | 40.00% |
> | CHORD | 11.40% | 31.25% |
> | PEAR | 12.50% | 42.50% |
> We commit to (1) adding a discussion on SFT/RL hybrid and clarify the scope differences, and (2) cite prominent works like CHORD, in the next revision.
>
>
> We include two new baselines (both cited in the paper): DFT and PSFT (Proximal-SFT). We report average acc. (avg@64, same as in the paper) across AIME-25,-24, and AMC23.
>
>
>
> | Method | DS-Qwen-1.5B Avg. | Qwen3-4B-Base Avg. |
> | ------ | ----------------: | -----------------: |
> | DFT | 17.33 | 19.00 |
> | PSFT | 24.67 | 15.67 |
> | PEAR | 29.00 | 28.33 |
>
>
> PEAR + GRPO yields significantly better performance. As discussed in Sec 4.3 of our paper, one needs to weigh the future, not the current token, to better prepare for RL.
>
> ---
>
>
> > **W4: Will PEAR hinder a model's learning of novel knowledge or formats?**
>
>
> Thanks for the discussion! This may happen to methods that downweight tokens solely because their own conditional probability is low, but PEAR assigns weights based on future plausibility.
>
>
> We have 3 pieces of additional evidence:
>
>
> (1) PEAR still learns useful reasoning patterns rather than collapsing them; (2) PEAR does not hurt learning of required answer formats (`\boxed{}`) (3) its gains transfer across domains, e.g., from offline SynLogic training to RL on Enigmata (Table 3 in the paper).
>
>
> ** Reasoning pattern count, full analysis on next revision**.
> | approach | wait | hmm | alternatively | okay | therefore |
> |----------|-----:|----:|--------------:|-----:|----------:|
> | SFT      | 1691 | 76  | 234           | 176  | 300       |
> | PEAR     | 1540 | 74  | 267           | 158  | 244       |
>
>
> ---

---

> > ### Author Rebuttal · Reviewer_Bcfh · 2026-04-01
> >
> > The rebuttal addresses several of my main concerns in a meaningful way. In particular, the additional hyperparameter sensitivity analysis reduces my concern that PEAR may rely on heavy dataset-specific tuning, and the newly added instruction-following results substantially strengthen the claim that the method is useful beyond pure math/logic settings. The added comparisons to CHORD/DFT/PSFT are also helpful and improve the empirical positioning of the work.
> >
> > That said, some concerns remain only partially resolved. The robustness evidence is still somewhat limited in scope, the applicability to standard RLHF settings with learned reward models remains unverified, and the discussion around potential suppression of useful low-probability trajectories is suggestive but not yet fully conclusive.
> >
> > Overall, the rebuttal increases my confidence in the paper, but does not fundamentally change my assessment of its scope and limitations. I therefore keep my score.

---

> > > ### Author Response · Authors · 2026-04-01
> > >
> > > Thank you for the thoughtful re-evaluation and for acknowledging that the rebuttal addressed several main concerns, including the hyperparameter sensitivity analysis, instruction-following results, and new comparisons!
> > >
> > > ---
> > >
> > > **Scope.** As noted in our abstract and throughout the paper, PEAR targets **verifiable reasoning**, where model responses are structured, search-like traces. That said, we invested considerable effort during the rebuttal to extend beyond this core scope: we trained full SFT ->RL pipelines on **instruction-following** tasks and evaluated them on IF-Eval and IF-Bench. We are glad the reviewer found these results strengthening the claim that PEAR generalizes beyond math/logic reasoning.
> > > ​
> > > We respectively clarify that extending PEAR to RLHF is beyond the intended scope of the submission but we agree this would be a natural next step to investigate and would be glad to discuss it further!
> > >
> > > ---
> > >
> > > **Hyperparameter sensitivity.** We want to emphasize that PEAR uses a single set of clipping hyperparameters across all experiments — these function as standard stabilization, not per-dataset tuning.
> > > In addition to the 4 clip-range variants in our rebuttal, we **now also provide a block-size ablation** (block-1/4/8/seq on Qwen3-1.7B-Base, SynLogic), which reveals a clear stability–granularity tradeoff – block level may introduce variance, while sequence level may lose fine-grained information (lower pass@8).
> > >
> > >
> > > | Setting | Pass@1 | Pass@8 |
> > > |---|---:|---:|
> > > | Seq | 57.50% | 67.50% |
> > > | block-4 | 60.00% | 83.75% |
> > > | block-8 | 68.75% | 77.50% |
> > > | block-1 | 45.00% | 70.00% |
> > >
> > > We welcome **specific additional hyperparameter studies** the reviewer has in mind, we would be glad to perform those analyses if resources permit.
> > >
> > > ---
> > >
> > > **Low-probability trajectory suppression.** Our rebuttal addressed this with both conceptual arguments (suffix-based weighting is self-correcting; PEAR reweights rather than discards) and empirical evidence (reasoning pattern counts confirming effective learning, accuracy gains confirming effective format learning, and cross-domain transfer in Table 4 of the original paper demonstrating that PEAR preserves useful structure).
> > >
> > > We would genuinely appreciate learning **what specific experiment or analysis the reviewer believes would be most informative here**, and we would be glad to pursue it if feasible!
> > >
> > > ---
> > >
> > > We are grateful for your careful and constructive reading of the paper! Also, given that the reviewer notes the rebuttal meaningfully addressed several main concerns, we would value understanding what specific additional evidence would warrant reconsidering the assessment. We remain happy to engage further during the discussion period!

---

### Official Review · Reviewer_9VYk · 2026-03-13

**Soundness:** 3
**Presentation:** 3
**Significance:** 3
**Originality:** 3
**Overall Recommendation:** 4
**Confidence:** 4

**Summary:**

This paper observes that optimizing SFT for peak offline accuracy can yield sub-optimal initializations for downstream RL, due to a distribution mismatch between the offline data and the model's own rollout distribution. To address this, the authors propose PEAR, which reweights the SFT loss via importance sampling between the current policy π_θ and the behavior policy π_β. A key design is the "suffix" weighting: the IS ratio at token t is computed over *future* tokens (t+1 to T), inspired by Off-Policy Evaluation, capturing how well the continuation aligns with the model's own distribution. Three granularity variants (token, block, sequence) are provided. Experiments on math reasoning benchmarks across 1.5B–8B models show that PEAR-initialized RL substantially outperforms standard SFT-initialized RL.

**Compliance With Llm Reviewing Policy:**

Affirmed.

**Final Justification:**

The rebuttals have resolved my questions. This paper offers an interesting perspective regarding a proper design of SFT loss to improve the overall posttraining stage. I am inclined to rate it as positive hence recommend acceptance.

**Key Questions For Authors:**

1. **Generalization beyond reasoning.** Does PEAR help when the RL reward comes from a learned reward model (e.g., in RLHF for chat/safety)? If not tested, what is your hypothesis for why it would or would not transfer?

2. **Hyperparameter guidance.** How sensitive are results to block size B? Can you provide ablations across at least two settings? Is there a principled rule for choosing between token/block/sequence granularity?

3. **RL algorithm generality.** Does PEAR's benefit persist with PPO or online DPO? If GRPO-specific, the scope of claims should be qualified.

**Limitations:**

The authors discuss the dependence on π_β and IS variance, which is appreciated. However, the discussion would benefit from explicitly acknowledging the narrow evaluation domain (math reasoning only) and the absence of hyperparameter sensitivity analysis. No societal impact concerns are raised, which is appropriate for this type of methodological contribution.

**Strengths And Weaknesses:**

Strengths.

1. **Well-motivated problem with strong empirical evidence.** The paper clearly demonstrates that "better SFT ≠ better RL initialization" through Figure 1 and systematic experiments, turning an anecdotal practitioner's observation into a well-documented phenomenon. The gradient cosine similarity and NSS diagnostics convincingly show *why* PEAR works — its offline gradients are better aligned with on-policy RL gradients.

2. **Practical and lightweight.** PEAR requires only log-probability computation—no additional model forward passes. This makes it straightforward to integrate into existing SFT pipelines with minimal overhead, which is important for real-world adoption.

3. **Creative use of OPE for SFT preparation.** The suffix-based IS weighting (computing ratios over future rather than past tokens) inspired by OPE is interesting and outperforms one-step variant.

4. **Thorough experimental coverage.** The method is validated across multiple model families (Qwen2.5, Qwen3, DeepSeek-Distill), scales (1.5B–8B), and benchmarks. The improvements are consistent and substantial, not marginal.

Weaknesses.

1. **Strong dependence on knowing π_β.** PEAR requires log-probabilities from the behavior policy. This is feasible for self-generated data but problematic for curated or third-party datasets where the data-generating model is unknown. The paper acknowledges this but provides no empirical study of degradation under approximate π_β, which would be the most common practical scenario.


2. **Insufficient hyperparameter analysis.** The paper uses fixed hyperparameters for PEAR throughout but does not systematically study sensitivity to these choices across tasks and scales. Given that IS variance grows with sequence length, guidance on when to prefer block-level vs. token-level vs. sequence-level weighting is also missing.

---

> ### Author Rebuttal · Authors · 2026-03-31
>
> **TL;DR**
>
> Thanks for positive rating and helpful discussion. We address each concern with additional results and will include those in the next revision:
>
> -  PEAR works with proxy behaviors and still outperform SFT
> - PEAR is stable across clipping ranges
> - Instruction-following experiments show that PEAR works beyond reasoning.
> - Block size ablations show a tradeoff between granularity and stability.
> - PEAR is compatible with other RL algorithms - e.g.DAPO.
> ---
>
> > W1: Effectiveness under approximate behavior policy.
> PEAR can be made applicable when the true behavior policy is inaccessible via 2 strategies:
> Single proxy behavior model to estimate behavior probabilities, and
> model ensemble, where the behavior probability is approximated by averaging probabilities across multiple proxies (Qwen series).
>
> All proxy variants substantially outperform **SFT + GRPO**. This anticipated performance drop from white-box version can be attributed to approximation errors in the behavior policy. Among the models we tested, **Qwen3-4B-Thinking** has the lowest Pearson correlation coefficient and correspondingly the weakest proxy performance.
>
> These practical alternatives enable PEAR to be operationalized even when the true behavior policy is unavailable.
>
> **Qwen3-1.7B-Base**
>
> | Method | Pass@1 | Pass@8 |
> |---|---:|---:|
> | SFT + GRPO | 15 | 45 |
> | PEAR (White Box) + GRPO | 45 | 70 |
> | PEAR (Qwen3-4B-Approx) | 38 | 69 |
> | PEAR (Qwen3-4B-Thinking-Approx) | 35 | 63 |
> | PEAR (Qwen3-32B-Approx) | 45 | 66 |
> | PEAR (Mixture) | 36 | 64 |
>
>
> | Proxy | Pearson wrt behavior |
> |---|---:|
> | Qwen3-32B | 0.7328 |
> | Qwen3-4B | 0.7237 |
> | Ensemble | 0.7263 |
> | Qwen3-4B-Thinking | 0.6605 |
>
> ---
>
> > W2: Hyperparameter Analysis
>
> We conducted hyperparameter analysis over clip min and clip max on per-token ratios for **Qwen3-1.7B-Base**. The clip range for the log importance ratio is:
>
> - Min: \{-0.08, -0.3\}
> - Max: \{0.08, 0.3\}
>
> Across these settings, we obtain **Pass@32 = 47.5% (std 3.95%)** and **Pass@64 = 52.19% (std 2.58%)** in the pre-RL setting. Hyper-parameter variation causes mild fluctuations, but does not lead to drastic behavioral changes.
>
>
> | Clip range | Pass@32 / Pass@64 |
> |---|---:|
> | [-0.08, 0.08] | 42.50 / 48.75 |
> | [-0.08, 0.3] | 46.25 / 52.50 |
> | [-0.3, 0.08] | 50.00 / 52.50 |
> | [-0.3, 0.3] | 51.25 / 55.00 |
>
> ---
>
> > Q1: PEAR’s applicability on non-reasoning domains.
>
> Thanks for the discussion. PEAR is initially motivated by the reasoning setup, which can be viewed as a form of in-context search, but demonstrating generalization beyond reasoning would indeed strengthen the work.
>
> We therefore add experiments on **instruction-following**. We perform SFT/PEAR on the instruction-following split of **SYNTHETIC-2** , and RL using **AI2’s Dolci-RL-Zero-IF-7B**. We use **Qwen3-0.6B** as the student model and observe a clear advantage for PEAR.
>
> | Model | IF-Eval: SFT + GRPO | IF-Eval: PEAR + GRPO | IF-Bench: SFT + GRPO | IF-Bench: PEAR + GRPO |
> |---|---:|---:|---:|---:|
> | Qwen3-0.6B-Base | 37.34% | 57.67% | 16.67% | 29.59% |
>
> We have not yet directly tested PEAR in a full learned-RM setting due to compute limitations. Our hypothesis is that PEAR should still help whenever there is an offline-to-online policy mismatch. We will include more experiments in the next revision if compute resources permit.
>
> ---
>
> > W2+Q2: Block size analysis.
>
> We study the effect of block size on **Qwen3-1.7B-Base** using **SynLogic** and observe a clear trade-off between stability and granularity, consistent with our discussion in the paper.
>
> At the fine-grained end, **block size 1** (token-level actions) yields lower post-RL **Pass@1** compared to **block-4** and **block-8**, suggesting lower training stability when importance weights are applied at the token level.
>
> At the coarse-grained end, **whole-sequence importance sampling** achieves the lowest **Pass@8** and also underperforms **block-4** and **block-8** on **Pass@1**.  This suggests the trade off between stability and fine-grained information.
>
>
> We see intermediate block sizes (e.g., **4** and **8**) provide a better balance, achieving higher performance by maintaining sufficient stability while preserving meaningful granularity.
>
> | Setting | Pass@1 | Pass@8 |
> |---|---:|---:|
> | Seq | 57.50% | 67.50% |
> | block-4 | 60.00% | 83.75% |
> | block-8 | 68.75% | 77.50% |
> | block-1 | 45.00% | 70.00% |
>
> ---
>
> > Q3: Compatibility with other RL algorithms.
>
> Thank you for raising this question.
>
> We conducted experiments on **SynLogic** with **DAPO**, which is widely used in RLVR. These results further show PEAR’s compatibility beyond GRPO. We train **Qwen3-1.7B-Base** on SynLogic under the same setup as in the paper, and find that **PEAR + DAPO** continues to outperform **SFT + DAPO**.
>
> We will include these results in the next revision, and experiment with other RL algorithms as compute permits.
>
> | Method | Pass@1 | Pass@8 |
> |---|---:|---:|
> | SFT + DAPO | 42.0% | 71.25% |
> | PEAR + DAPO | 60.00% | 80.00% |

---

> > ### Author Rebuttal · Reviewer_9VYk · 2026-04-07
> >
> > Thank you for your responses! Hope you could integrate the new resulst into a polished paper to make it more solid. I am inclined to keep my positive score and good luck for your submission!

---

> > > ### Author Response · Authors · 2026-04-07
> > >
> > > Thank you for your encouraging feedback and for confirming that your concerns have been **fully resolved** — we sincerely appreciate the time and effort you dedicated to improving our work! We will integrate all new results and analyses into the next revision to further strengthen the paper as suggested.
> > >
> > > We are happy to engage in further discussions to help clarify our work!

---

### Official Review · Reviewer_7PYF · 2026-03-13

**Soundness:** 3
**Presentation:** 3
**Significance:** 3
**Originality:** 3
**Overall Recommendation:** 4
**Confidence:** 4

**Summary:**

The paper makes an observation that for post-training, good SFT performance is not indicative of the good performance after RL. In fact, the opposite happens where good SFT performance leads to worse performance after RL. To mitigate this problem, the authors propose PEAR which uses importance sampling ratios as a weight during SFT which prepares the model for RL. This leads to improved performance when the model is post-trained using reinforcement learning.

**Compliance With Llm Reviewing Policy:**

Affirmed.

**Final Justification:**

My concerns have been addressed

**Key Questions For Authors:**

I have some key concerns:

1. This technique will only work if we have access to the behavioral policy that generated the SFT data. This might not hold in many cases, for instance cases where the data is scraped from the web or white-box access is not available.
2. I think the paper could be strengthened by a simple baseline which is, early stopping in SFT based on coverage estimated using pass@K where K could be a high enough number such as 32/128.
3. I would also like the authors to answer this: if you are assuming access to a behavioral policy then why can't you perform on-policy distillation [1, 2] instead of a separate SFT and RL stage?

[1] On-Policy Distillation of Language Models: Learning from Self-Generated Mistakes (https://arxiv.org/abs/2306.13649)
[2] https://thinkingmachines.ai/blog/on-policy-distillation/

**Limitations:**

yes

**Strengths And Weaknesses:**

1. The paper is comprehensive in its study of various alternative to log-likelihood in supervised training and outperforms them in their set of proposed methods.
2. The method proposed is easy to implement and is principled, using ideas such as off-policy evaluation to come up with the objective.
3. The experiments are done on a variety of settings which adds to the strengths of the paper.

---

> ### Author Rebuttal · Authors · 2026-03-31
>
> **TLDR**:
> - **White-box assumption**: Our new experiments show that PEAR does not require exact white-box access to the behavior policy in order to be effective: proxy behavior models can still work well in practice.
> - **Offline Pass@Large K Criterion**: We appreciate the reviewer’s suggestion of using pass@32/128 as a model-selection criterion. In fact, this is a signal we already examined in the original submission, and our results suggest that pass@large K (e.g., pass@128) is not a reliable predictor of downstream post-RL performance (Figure 1, panel 3 in the submission).
> - **OPD vs SFT-RL Pipeline**: 1) we show PEAR can work without white box access. 2) OPD aims to mimic teacher’s behavior, SFT-RL pipeline is not bounded by any teacher.
>
> ---
>
> > Q1. Can PEAR work with Black-box scenario?
>
> Exact access to the behavior policy is not necessary in practice. When the exact behavior policy is inaccessible, we propose two practical strategies to make PEAR applicable in real settings: (1) using a different proxy behavior model to estimate the behavior probabilities, and (2) using a model ensemble, in our case multiple Qwen3-series proxy models (-4B,-4B-thinking, 14B) excluding behavior generator, where the behavior probability is approximated by averaging probabilities across multiple models. We experiment on the same synthetic logic game setting as in the paper.
>
>
> We observe that all proxy variants substantially outperform SFT + GRPO even though it may underperform white-box. The performance drop is expected and can be attributed to the errors in behavior approximation. We also computed Pearson correlation between proxy's token probabilities and the actual behavior model across the dataset. Among the experimented models, we saw Qwen3-4B-Thinking has lowest Pearson correlation coefficient and therefore weakest proxy performance.
>
> These practical alternatives enable PEAR to be operationalized even when the true behavior policy is unavailable.
>
>
> **Qwen3-1.7B-Base**
> | Method                          | Pass@1 | Pass@8 |
> | ------------------------------- | -----: | -----: |
> | SFT+GRPO                        |     15 |     45 |
> | PEAR (White Box) + GRPO         |     45 |     70 |
> | PEAR (Qwen3-4B-Approx)          |     38 |     69 |
> | PEAR (Qwen3-4B-Thinking-Approx) |     35 |     63 |
> | PEAR (Qwen3-32B-Approx)         |     45 |     66 |
> | PEAR (Mixture)                  |     36 |     64 |
>
> **Qwen3-4B-Base**
> | Method                  | Pass@1 | Pass@8 |
> | ----------------------- | -----: | -----: |
> | SFT+GRPO                |     43 |     71 |
> | PEAR (White Box) + GRPO |     64 |     76 |
> | PEAR (Qwen3-32B-Approx) |     59 |     78 |
>
> **Pearson Correlation**
> | Model             | Pearson |
> | ----------------- | ------: |
> | Qwen3-32B         |  0.7328 |
> | Qwen3-4B          |  0.7237 |
> | Ensemble          |  0.7263 |
> | Qwen3-4B-Thinking |  0.6605 |
>
> ---
>
> > Q2: Why not On-policy distillation instead of SFT + RL when behavioral policy is accessible?
>
> OPD is indeed a regime of clear and growing interest in the community, and we view it as an important complementary direction to the SFT + RL pipeline instead of a substitution. OPD’s goal is to approximate a teacher model’s behavior,  whereas in SFT + RL pipeline the final policy is not constrained to remain an imitation of a teacher policy, since the RL stage further optimizes the model on its own sampled rollouts. Built upon this premise, PEAR aims to produce better initializations for the RL stage by proposing improvements to SFT objectives. Importantly, our black box experiments show PEAR can lift the assumption of behavior policy full access.
>
> ---
>
> > Q3: Pass@large K as a model selection criterion.
>
> We thank the reviewer for suggesting early stopping using offline proxy pass@K! This is an intuitively appealing candidate criterion for model selection and we therefore carefully investigated it in the paper and concluded it is not a reliable signal for model selection.  In the original paper, compared the offline pass@128’s among multiple objectives (panel 3 on Figure 1 in the paper) and their post-RL performances, and show models yielding higher offline pass@Large K can still underperform after identical RL training.
>
> ---
>
> We hope our additional results and discussion above can address your questions, and we look forward to additional discussions!

---

> > ### Author Rebuttal · Reviewer_7PYF · 2026-04-04
> >
> > My concerns have been addressed. I will increase my score

---

> > > ### Author Response · Authors · 2026-04-04
> > >
> > > Thank you for your response — we are glad to hear that our rebuttal helped address your concerns, and we sincerely appreciate your re-evaluation of our work. We would be very happy to continue the discussion if there are any remaining questions or aspects that we could further clarify to strengthen the paper!

---

### Decision · Program_Chairs · 2026-04-30

**Decision:**

Accept (regular)

**Comment:**

The paper investigates a key issue in the SFT→RL post-training pipeline for LLMs, showing that stronger SFT performance can lead to worse downstream RL outcomes due to distribution mismatch. To address this, the authors propose PEAR, an importance-sampling-based reweighting method applied during SFT to better align with the RL objective. The method is simple, principled, and evaluated across multiple models and reasoning benchmarks, demonstrating consistent improvements after RL.

Reviewers agree that the problem is well-motivated and important, and find the method practical, easy to implement, and empirically strong. The experimental validation is considered thorough, and the core insight is viewed as valuable for the community. Main concerns include reliance on access to the behavior policy, limited evaluation scope beyond reasoning tasks, and the need for more analysis on hyperparameters and comparisons with related methods.

During rebuttal, the authors provided substantial additional experiments and clarifications, including results with proxy behavior policies, hyperparameter sensitivity analysis, broader evaluation on instruction-following tasks, and additional baselines. Most reviewers indicated that their concerns were fully addressed, while one reviewer noted improvements but retained some reservations about scope.

Overall, given the positive reviewer consensus and strengthened empirical support after rebuttal, I recommend acceptance.